# A putative cap binding protein and the methyl phosphate capping enzyme Bin3/MePCE function in telomerase biogenesis

Diego J. Páez-Moscoso[1,2,4], David V. Ho [1,2], Lili Pan [1], Katie Hildebrand[2,5], Kristi L. Jensen [1], Michaella J. Levy[2,6], Laurence Florens [2] & Peter Baumann [1,2,3✉]

Telomerase reverse transcriptase (TERT) and the noncoding telomerase RNA (TR) subunit constitute the core of telomerase. Additional subunits are required for ribonucleoprotein complex assembly and in some cases remain stably associated with the active holoenzyme. Pof8, a member of the LARP7 protein family is such a constitutive component of telomerase in fission yeast. Using affinity purification of Pof8, we have identified two previously uncharacterized proteins that form a complex with Pof8 and participate in telomerase biogenesis. Both proteins participate in ribonucleoprotein complex assembly and are required for wildtype telomerase activity and telomere length maintenance. One factor we named Thc1 (Telomerase Holoenzyme Component 1) shares structural similarity with the nuclear cap binding complex and the poly-adenosine ribonuclease (PARN), the other is the ortholog of the methyl phosphate capping enzyme (Bin3/MePCE) in metazoans and was named Bmc1 (Bin3/MePCE 1) to reflect its evolutionary roots. Thc1 and Bmc1 function together with Pof8 in recognizing correctly folded telomerase RNA and promoting the recruitment of the Lsm2-8 complex and the catalytic subunit to assemble functional telomerase.

[1] Faculty of Biology, Johannes Gutenberg University, 55099 Mainz, Germany. [2] Stowers Institute for Medical Research, Kansas City, MO 64110, USA.
[3] Institute of Molecular Biology, Ackermannweg 4, 55128 Mainz, Germany. [4] Present address: Institute of Molecular Biology, Ackermannweg, 4 55128 Mainz, Germany. [5] Present address: Transgenic and Gene-Targeting Institutional Facility, University of Kansas Medical Center, 3901 Rainbow Boulevard, Kansas City, KS 66160, USA. [6] Present address: KCAS, 12400 Shawnee Mission Parkway, Shawnee, KS 66216, USA. ✉email: peter@baumannlab.org

Telomeres are protective caps at the ends of eukaryotic linear chromosomes that prevent natural DNA ends from being recognized as double-strand breaks. Deficiency in maintaining a functional telomere structure results in cellular senescence and chromosome instability that lead to premature aging and cancer[1]. Central to telomere maintenance is the extension of shortened telomeres by telomerase, a specialized reverse transcriptase that utilizes part of a long non-coding RNA subunit as the template for the synthesis of telomeric DNA repeats[2]. Telomerase is comprised of the catalytic protein Telomerase reverse transcriptase (TERT, Trt1 in *Schizosaccharomyces pombe*), the telomerase RNA subunit (TR, TER1 in *S. pombe*), and an unknown number of other protein subunits involved in the biogenesis, regulation, and recruitment of the enzyme to telomeres. Mutations in most known telomerase components have been implicated in premature aging diseases, such as dyskeratosis congenita, aplastic anemia, idiopathic pulmonary fibrosis, Hoyeraal–Hreidersson syndrome, Coats Plus syndrome, and Revesz syndrome (reviewed in ref. [3]). Reactivation or upregulation of telomerase activity is observed in over 85% of cancers. Given the critical role of telomerase in controlling cell proliferation, it is not surprising that telomerase ribonucleoprotein particle (RNP) biogenesis is a tightly regulated hierarchical assembly process involving numerous factors and stages[4,5].

In *S. pombe*, the telomerase RNA subunit TER1[6,7] is transcribed by RNA polymerase II and the primary transcript contains two exons and an intron[8]. This precursor is processed by the spliceosome. Deviating from a normal splicing reaction, the intermediates generated by the first spliceosomal cleavage reaction are released with the 5′ exon being further processed into mature telomerase RNA and the intron lariat-exon 2 being degraded. This "discard" of splicing intermediates[9] is promoted by RNA elements within TER1 that are unfavorable for a transition to the second step of splicing. Interestingly, the spliceosome also functions as a 3′ end processing machinery for telomerase RNA in other species, but the underlying mechanisms for releasing splicing intermediates as cleavage products are remarkably diverse[10,11].

Prior to spliceosomal cleavage, the hetero-heptameric Sm complex binds TER1 directly upstream of the 5′ splice site and recruits the methyl transferase Tgs1, which hypermethylates the 7-monomethylguanosine cap at the 5′ end of TER1[12]. Following spliceosomal cleavage, the Sm complex is replaced by the Sm-Like complex Lsm2-8, which protects TER1 from degradation by exonucleases like the exosome[12–14]. Importantly, the binding of Lsm2-8 to telomerase is also a prerequisite for the binding of the catalytic subunit Trt1 to assemble an active enzyme[12].

Recent genetic and biochemical studies identified the Pof8 protein as a new factor involved in telomerase biogenesis[15–17]. Originally described as a putative F-box protein[18], profile database searches revealed structural similarities between Pof8 and the La-related protein (LARP) super family[15–17]. The La protein and La-related proteins bind to diverse noncoding RNAs and participate in RNA processing and RNP assembly[19]. Genuine La protein recognizes the 3′ oligo-uridine tail present on most nascent RNA polymerase III transcripts to protect them and in many cases influences the folding of the RNA into functional conformations[20,21]. Based on sequence conservation and domain organization, the LARP family has been divided into four subfamilies (LARP1, 4, 6, and 7). Pof8 shows the most similarity with LARP7 proteins. Members of this family have predominantly been studied in the context of the 7SK complex[22]. However, recently additional roles in the context of other RNPs have been described, including a function for LARP7 as a molecular bridge between U6 snRNA and a subset of box C/D small nucleolar RNAs that guide U6 2′-O-methylation[23,24].

In the ciliated protozoa *Tetrahymena thermophila* and *Euplotes aediculatus*, the respective LARP family proteins p65 and p43 are bona fide telomerase subunits associated with the active enzymes[25,26]. Further characterization of *Tetrahymena* p65 revealed its function as RNA chaperone in the hierarchical assembly of telomerase, whereby p65 induces a conformational change in the RNA that is favorable for the binding of TERT[27,28]. The human La and Larp7 proteins have also been implicated in telomere maintenance:[29,30] Larp7 loss-of-function mutations in patients with Alazami syndrome being associated with very short telomeres and Larp7 knockdown causing a reduction in telomerase activity in cancer cells. Whether human Larp7 is directly associated with telomerase RNA and functions in an analogous way to p65 has remained unclear.

The fission yeast Pof8 protein binds to telomerase RNA and promotes the association of the Lsm2-8 complex with TER1, which in turn promotes the binding of the catalytic subunit Trt1[12,15–17]. Loss of Pof8 reduces TER1 stability, largely compromises telomerase assembly, and results in critically short telomeres[15–17]. Structurally, Pof8 is comprised of a conserved N-terminal La module (La motif plus RNA recognition motif 1 (RRM1)) and a C-terminal RRM2. The conservation of the RNP3 motif (Y/W-X-D/Q) and the position of the α3 helix, as revealed by the crystal structure and NMR studies of the C-terminal domain, identifies RRM2 as an xRRM despite the absence of an extended α3 helix[31]. This places Pof8 structurally as well as functionally in the same family as p65 and LARP7. The loss of stability of the holoenzyme in the absence of Pof8[15] and the observation that Pof8 is associated with nearly 100% of telomerase enzyme[17] support a role in stabilizing an active conformation of the holoenzyme, rather than functioning solely as assembly chaperon.

The biochemical and structural basis for the constitutive presence of Pof8 on telomerase was elegantly explained by a recent study that revealed direct interactions between Pof8 and several subunits of the Lsm complex[32]. The authors showed that Pof8 and Lsm bind telomerase RNA cooperatively, with Pof8 specifically recognizing a correctly folded pseudoknot and Lsm2-8 interacting with the oligo-uridine tail at the 3′ end of the RNA. Extensive protein–RNA contacts and cooperative binding ensure that correctly folded TER1 is protected and then licensed for assembly with the catalytic subunit.

As LARP family proteins often function in complex with other factors, we wondered whether Pof8 has binding partners in addition to known telomerase components. Here we show that Thc1, a previously uncharacterized open reading frame, and Bmc1, the putative fission yeast ortholog of metazoan methyl phosphate capping enzymes, form a complex with Pof8, revealing an intriguing compositional similarity between telomerase and the 7SK RNP in metazoans. Deletion of either *thc1* or *bmc1* results in decreased TER1 steady-state levels, telomerase assembly defects, and shortened telomere length. In aggregate, our findings provide further insights into telomerase biogenesis as a hierarchical assembly process. The structural similarity of Thc1 with human PARN provides an evolutionary link in the biogenesis of telomerase in distant organisms, whereas Bmc1 functioning in telomerase biogenesis adds to the plethora of recently discovered interconnections between functionally diverse non-coding RNAs, including snRNAs, snoRNAs, RNaseP, and now 7SK RNA.

## Results

**Pof8 forms a complex with Thc1 and Bmc1**. To gain further insight into how Pof8 mediates the loading of the Lsm2-8 complex onto TER1, we immunoprecipitated epitope-tagged Pof8 expressed under the control of its endogenous promoter. Silver staining of the immunoprecipitated proteins revealed enrichment of four bands in

the tagged samples over untagged controls (marked with an asterisk in Supplementary Fig. 1a). The samples were then analyzed by multidimensional protein identification technology (MudPIT) and 19 proteins with an average distributed Normalized Spectral Abundance Factor (dNSAF) of >0.005 in the tagged samples were found to be enriched in the Pof8 immunoprecipitates (Supplementary Fig. 1b and Supplementary Table 1). Among them were Pof8 itself and Lsm4, a member of the Lsm2-8 complex, which had previously been shown to directly interact with Pof8 based on crosslinking experiments[32]. Four additional members of the Lsm2-8 complex were also enriched in the Pof8 pulldown but did not meet our stringent cutoff criteria. Based on their molecular weights, the two most prominently enriched proteins are Pof8 itself (49 kDa) and the heat-shock protein Ssa2 (70 kDa). Ssa2 is an abundant chaperon that frequently associates with epitope-tagged proteins and was also present in lower amounts in the untagged control immunoprecipitation (IP). We therefore considered its enrichment in the Pof8 IP a consequence of tagging Pof8 and did not further investigate it in this study. Two proteins caught our attention as they were only detected in the tagged Pof8 samples: sequence orphan SPCC18b5.09c and SPBC2A9.10, a predicted transcriptional and translational regulator of the Bin3 family. Their respective sizes are consistent with the two lower bands marked with asterisks in Supplementary Fig. 1a.

Sequence orphan SPCC18b5.09c encodes for a 13.5 kDa protein with no reported sequence homology to previously characterized proteins. However, a profile sequence search on the HHpred[33] server revealed similarity with the nuclear cap-binding protein 3 (NCBP3) and the poly(A)-specific ribonuclease (PARN) (Fig. 1a, b). The nuclear cap-binding complex (CBP20 and 80) and PARN have previously been implicated in the biogenesis of human telomerase[34–38], raising the possibility that SPCC18b5.09c may carry out an analogous function in fission yeast. Based on our results detailed below, we will subsequently refer to SPCC18b5.09c as Thc1 (Telomerase holoenzyme component 1).

The open reading frame SPBC2A9.10 shares sequence similarity with proteins in the MePCE/Bin3 family (Fig. 1c). As the name Bin3 is ambiguous, referring to Bicoid interacting protein 3, first identified in *Drosophila* and characterized by the presence of an AdoMet binding domain, as well as the unrelated Bridging-Integrator 3 protein, members of the former family are now preferentially referred to as Methyl phosphate capping enzymes (MePCE). In accordance with the nomenclature conventions for fission yeast genes and to maintain the link to the founding members of the family, *Drosophila* Bin3 and MePCE, we will refer to SPBC2A9.10 as Bmc1 (Bin3/MePCE 1). The RNA methyltransferase activity of mammalian MePCE adds a methyl group to the γ-phosphate of the 5′ cap of RNA polymerase III transcribed 7SK RNA, a major transcriptional regulator, and U6 snRNA, a component of the spliceosome[39]. The presence of a MePCE ortholog in fission yeast is puzzling since the 7SK RNA has not been identified in any yeast species and an orthologous protein is absent from the *Saccharomyces cerevisiae* genome, indicating that U6 snRNA can be stabilized in ways not involving a MePCE ortholog. As high throughput knockout studies reported *S. pombe* bmc1⁺ (SPBC2A9.10) as essential for viability[40,41], its target RNAs if any have remained uncharacterized thus far. Based on the known enzymatic activity, polIII transcripts such as U6 snRNA are prime candidates. However, we wondered whether the interactions of Bmc1 with Pof8 is indicative of involvement in the biogenesis of telomerase, despite the absence of a 5′ triphosphate cap structure on TER1.

**Reduced TER1 level and short telomeres in *thc1Δ* and *bmc1Δ* cells**. We next deleted *thc1⁺* in haploid cells and one copy of *bmc1⁺*

in diploid cells. Strains deleted for *thc1* were readily obtained. Surprisingly, sporulation of *bmc1⁺/Δ* cells also gave rise to healthy haploid colonies. The gene deletions were verified by polymerase chain reactions (PCRs; Supplementary Fig. 2a–d) and further confirmed by the absence of reads mapping to the respective loci in RNA sequencing experiments (Supplementary Fig. 2e, f). It is not clear to us why prior attempts to delete *bmc1* have not yielded viable knockout cells[40,41]. While we cannot rule out the possibility of a spontaneously arising suppressor mutation in our strains, backcrossing *bmc1Δ* strains to wild-type *S. pombe* obtained from the American Type Culture Collection (Lindner 972; ATCC#38366) followed by tetrad dissections yielded four viable colonies for 10 out of 14 tetrads, as well as the expected two-to-two segregation of the selection marker (Supplementary Fig. 2g).

To confirm the physical interactions suggested by the mass spectrometry result, we introduced an N-terminal epitope tag on Pof8 and C-terminal epitope tags on Thc1 and Bmc1. Following IP of Pof8, Thc1 (Fig. 2a) and Bmc1 (Fig. 2b) were readily detected. Since Pof8 has been shown to directly bind TER1[32] and structure-function predictions were consistent with Thc1 and Bmc1 also interacting with RNA, we tested whether the Pof8-Thc1 and Pof8-Bmc1 interactions are nucleic acid-dependent. The interactions were neither impaired by treatment with RNase A nor Benzonase, indicating that the interactions are not mediated by TER1 (Fig. 2a, b). Similarly, Thc1 coprecipitated with Bmc1 in a non-nucleic acid-dependent manner (Fig. 2c). In summary, these experiments confirmed the associations indicated by the mass spectrometry and suggested that Pof8, Bmc1, and Thc1 are part of a protein complex (Fig. 2d). As the IPs were caried out in *S. pombe* cell-free extract, it remains possible that interactions are mediated by additional proteins.

Based on the known function of Pof8, we next assessed whether Thc1 and Bmc1 also function in telomerase biogenesis. Examination of the steady-state levels of TER1 in cells deleted for Thc1 by reverse transcriptase–quantitative PCR (RT-qPCR) revealed a threefold reduction in TER1 levels (Fig. 3a), similar to the reduction we and others had previously observed in cells deleted for *pof8*[15–17]. Deletion of *bmc1* also resulted in a reduction of total TER1 levels, albeit not quite to the same extent as the other single gene deletions (Fig. 3a). Detection of TER1 with primers annealing to the first exon captures all forms of the RNA, with approximately 90% of the signal coming from the mature form based on northern analysis[6,9]. As the intron and second exon are removed by spliceosomal cleavage during the maturation process, their presence is specific to the precursor forms. Consistent with a role for the complex at the later stages of biogenesis, no reduction of the precursor was observed in any of the deletion mutants using a primer pair in exon 2 (Fig. 3b).

We next examined whether Thc1 and Bmc1 are stably associated with TER1 as previously reported for Pof8. TER1 was approximately 8000-fold enriched in immunoprecipitates of epitope-tagged Thc1 and Pof8 relative to untagged controls (Fig. 3c). A smaller, 64-fold-enrichment was observed with Bmc1, indicating a weaker interaction that is more easily disrupted under the experimental conditions. Alternatively, Bmc1 may not be a constitutive member of the telomerase complex and may only be associated with a fraction of the telomerase complex. Since telomere length regulation is exquisitely sensitive to changes in telomerase activity levels, we measured telomere length in the deletion strains. Telomeres were approximately 30% shorter than wild type in *thc1Δ* cells and 20% shorter in *bmc1Δ* cells (Fig. 3d). The reduction in telomere length in the mutants was observed across multiple isolates and was stable over the course of successive restreaks (Supplementary Fig. 3). Overall, these results confirm roles for Thc1 and Bmc1 in telomere maintenance and suggest functions in telomerase biogenesis or quality control.

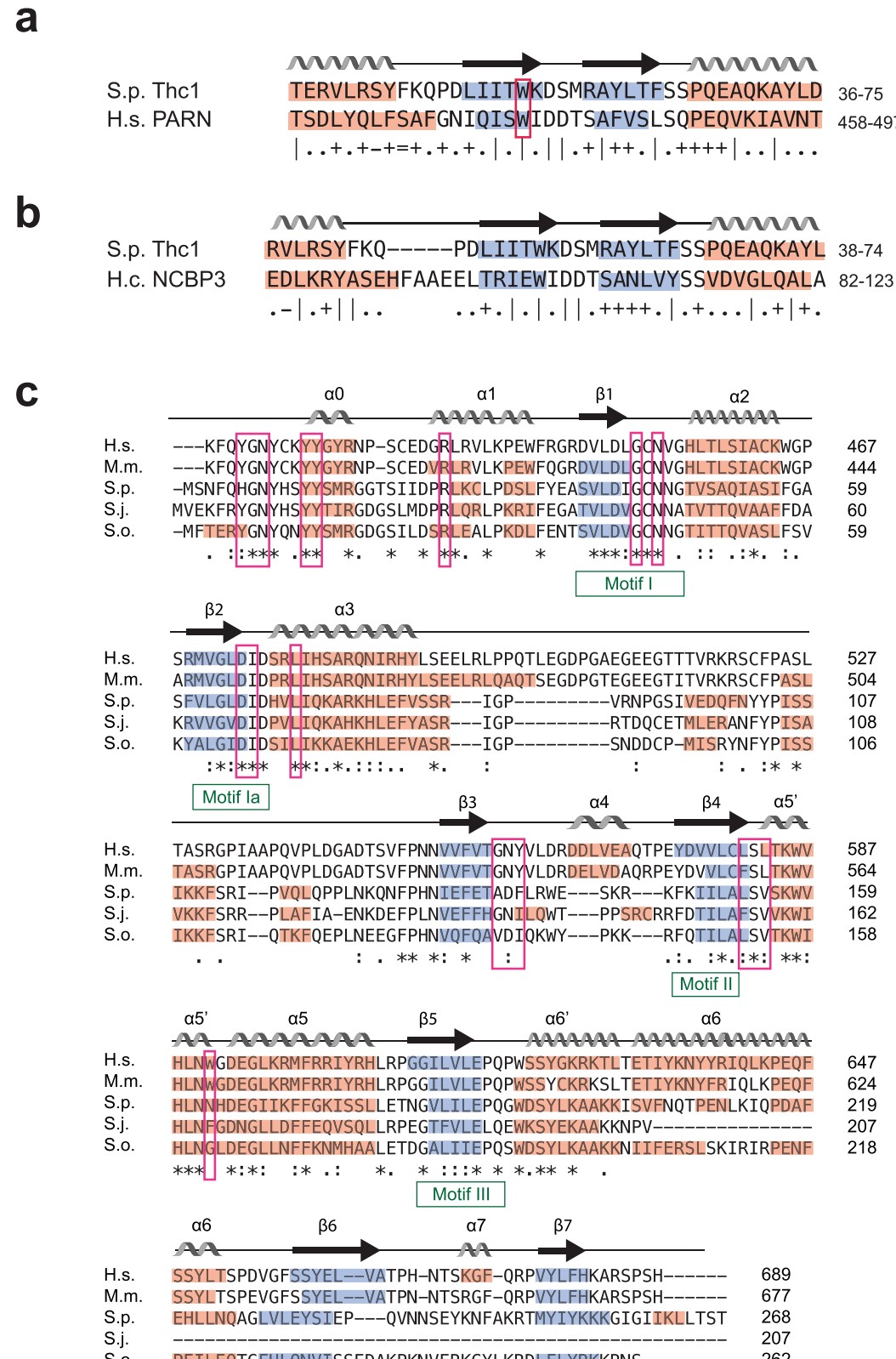

**Bmc1 function in telomerase biogenesis is independent of its catalytic activity**. As vertebrate MePCE and *Drosophila* Bin3 bind the 5′ triphosphate caps on 7SK and U6 snRNA and function enzymatically by transferring a methyl group onto the γ-phosphate, it is not apparent how TER1 with its trimethylguanosine cap could be targeted by Bmc1. Nevertheless, sequence alignments and homology modeling showed that the amino acids involved in AdoMet binding and methyltransferase activity are largely conserved in fission yeast (boxed in red in Fig. 1c). Furthermore, an overlay[43] of the AlphaFold[44] predicted structure for Bmc1 with the human MePCE structure[45] confirmed striking structural similarities consistent with a methyltransferase activity

**Fig. 1 Thc1 and Bmc1 share sequence and structural similarities with known RNA-processing factors. a** HHPred predictions for Thc1 reveal a putative cap-binding domain. Alignment and secondary structure elements of a putative cap-binding domain in Thc1 based on alignment with human (H.s.) poly-adenosine-specific ribonuclease (PARN); b-strands are highlighted in blue and a-helices in red. The secondary structure elements predicted for Thc1 are labeled above the alignment. The conserved tryptophane that stacks against the guanosine base in the PARN structure is boxed in red. Symbols indicating the quality of the alignment match are depicted below the sequence: "|" very good, "+" good, "." Neutral, "−" bad, and "=" very bad. **b** Alignment of the same region of Thc1 as in **a** with *Histoplasma capsulatum* (H.c.) Nuclear Cap Binding Protein 3 (NCBP3). **c** Multiple sequence alignment of human (H.s.) MePCE (amino acids 412–689), mouse (M.m.) MePCE (aa 395–677), *S. pombe* (S.p.) Bmc1, and putative Bmc1 orthologs from *S. japonicus* (S.j.) and *S. octosporus* (S.o.) using Clustal Omega. Secondary structure elements are annotated based on H.s. MePCE[45]. Motifs characteristic of SAM-dependent methyltransferases are indicated in green[47], residues involved in SAH binding by human MePCE are boxed in pink[45]. The following symbols are used below the alignment: "*" = identical amino acid across all sequences, ":" = amino acid groups of similar properties, "." = amino acid groups of weakly similar properties.

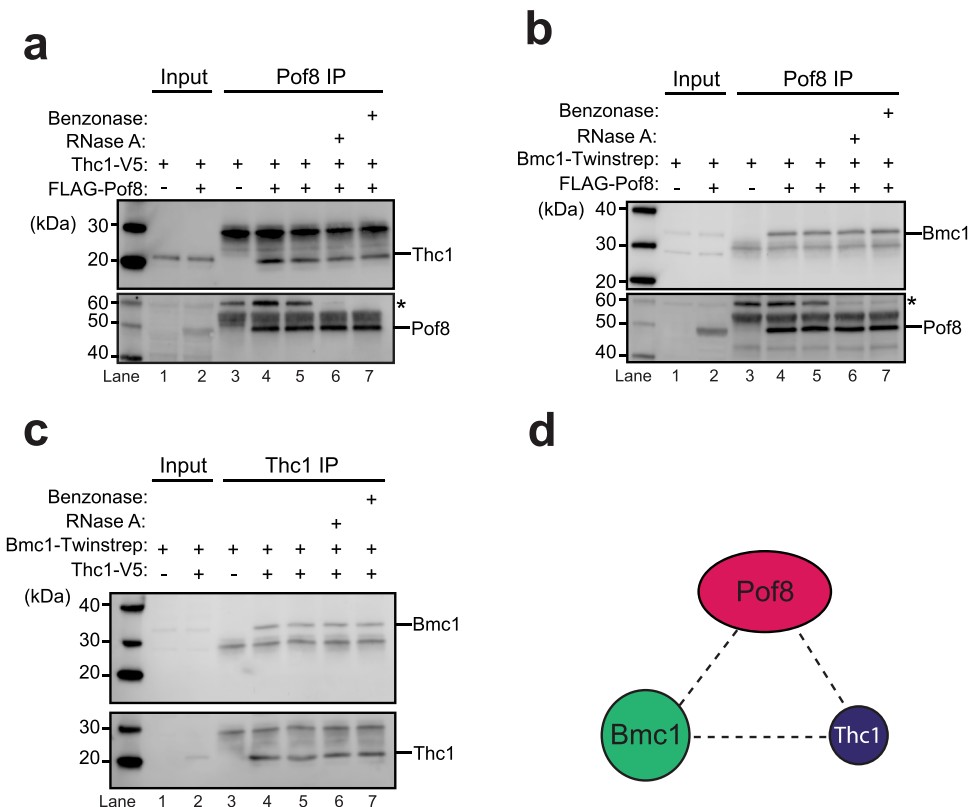

**Fig. 2 Associations between Pof8, Thc1, and Bmc1. a** Thc1 co-immunoprecipitates with Pof8. Co-IP of Thc1-2xV5 with 3xFLAG-tagged or untagged Pof8 by anti-FLAG. Western blot for input (10%, lanes 1 and 2) and IP (lanes 3–7) probed with anti-V5 to detect Thc1-2xV5 (*top*) and reprobed with anti-FLAG for 3xFLAG-Pof8 (*bottom*). Samples in lanes 5–7 were incubated at 37 °C for 15 min to assess whether the interactions are mediated by nucleic acid. An asterisk marks a protein of approximately 60 kDa that is recognized by the FLAG antibody even in cells that lack FLAG-tagged Pof8 (lane 3). This 60 kDa protein of unknown identity is enriched in the immunoprecipitate in an RNA-dependent, but FLAG-Pof8 independent manner. **b** Bmc1 co-immunoprecipitates with Pof8. IP with anti-FLAG in extracts from strains harboring Twinstrep-tagged Bmc1 and 3xFLAG-tagged or untagged Pof8. Plus indicates the presence of the tag. Western blot for input (10%, lanes 1 and 2) and IP (lanes 3–7) probed with anti-strep-tag II for Bmc1-Twinstrep (top) and reprobed with anti-FLAG for 3xFLAG-Pof8 (bottom). Samples in lanes 5–7 were treated as in **a**. Asterisk marks the same non-specific band as described in **a**. **c** Bmc1 co-immunoprecipitates with Thc1. Co-IP of Twinstrep-tagged Bmc1 with 2xV5-tagged or untagged Thc1. Western blot for input (10%, lanes 1 and 2) and immunoprecipitated (lanes 3–7) probed with anti-strep-tag II for Bmc1-Twinstrep (top) and anti-V5 for Thc1-2xV5 (bottom). Samples in lanes 5–7 were treated the same way as in **a**. **d** Schematic of interactions among the three proteins.

for Bmc1 (Supplementary Fig. 4a). We therefore generated several mutants designed to abolish the putative capping activity based on structural and functional information on the orthologous proteins[45–47] and introduced them on plasmids under the control of the endogenous promoter into *S. pombe* cells deleted for endogenous *bmc1*. Assessment of protein expression levels showed that F246A, GCN43ACA, and EPQ188APA were expressed at similar levels to wild-type Bmc1, whereas expression of VLD39AAA was approximately threefold reduced (Supplementary Fig. 4b). Using telomere length as the terminal readout

of telomerase function in vivo, we found that each of the catalytic mutants rescued the telomere length phenotype, except for VLD39AAA (Supplementary Fig. 4c). As the expression of this mutant is reduced compared to wild type and the other mutants, it is likely that mutation of the VLD motif causes structural changes that compromise the stability of Bmc1. In summary, we surmise that the catalytic activity of Bmc1 is not required for its function in telomerase biogenesis. However, we cannot exclude the possibility that the mutants have residual catalytic activity, and a definitive conclusion will therefore require future

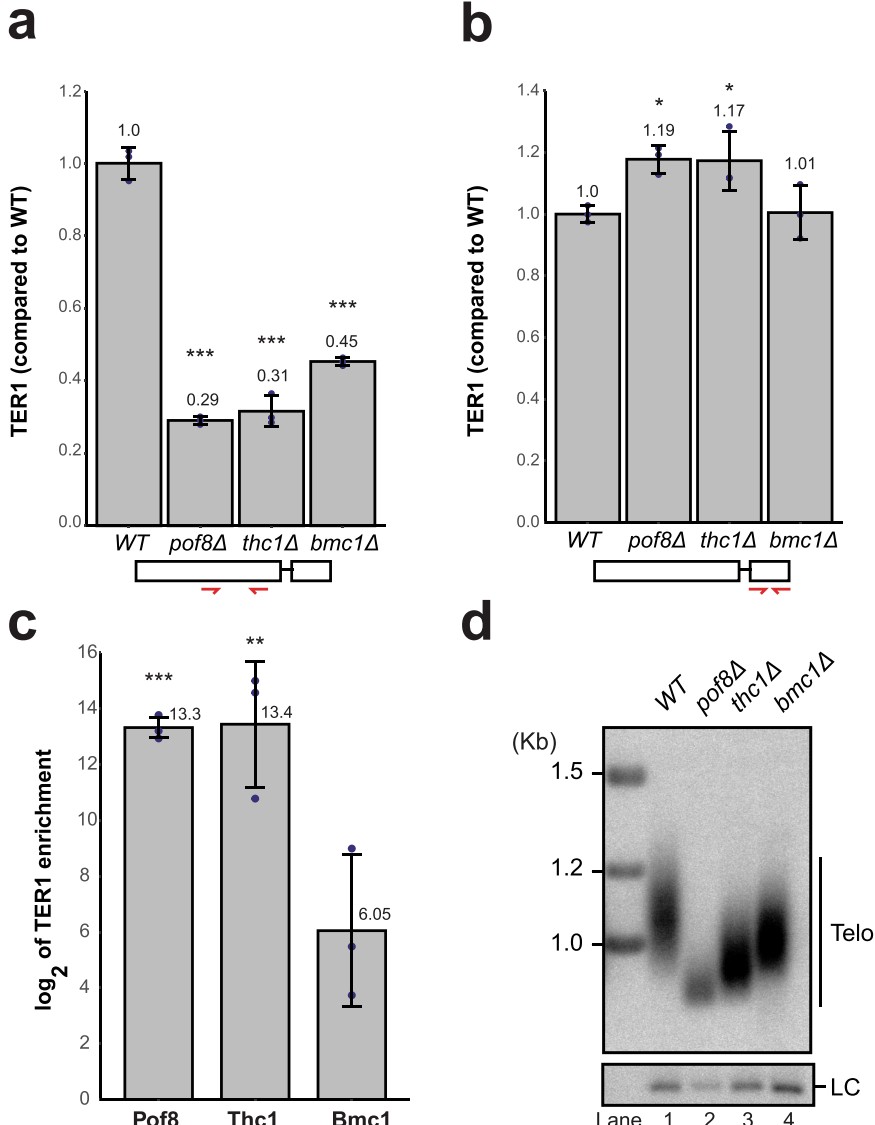

**Fig. 3 Thc1 and Bmc1 stabilize TER1 and are required for wild-type telomere length maintenance. a** RT-qPCR quantification of TER1 by from DNase-treated total RNA extracted from wild type (WT), *pof8Δ*, *thc1Δ*, and *bmc1Δ* strains. TER1 was normalized to three endogenous control genes (*act1*, *his1*, and *snR101*) and is shown relative to expression in WT. The bars show the mean (±SD, $n = 3$). Significant values are marked above each bar (two-sided Dunnett's test; ***$p < 0.001$, **$0.001 \leq p < 0.01$, *$0.01 \leq p < 0.05$). The schematic on the bottom indicates the position of the primers (red) used for TER1 amplification. **b** RT-qPCR quantification and statistical analysis as in **a** but using a primer pair in exon 2 of TER1, which is present in the precursor but not in the mature form of telomerase RNA. **c** RT-qPCR quantification of TER1 recovered from immunoprecipitations for 3xFLAG-Pof8, Thc1-2xV5, and Bmc1-Twinstrep normalized to untagged controls. The bars show the mean (±SD, $n = 3$) of enrichment of TER1 RNA relative to untagged control after normalization to RNA levels in the input using the same primer sets as in **a**. Statistical analysis as in **a**. **d** Telomeric Southern blot using genomic DNA from WT, *pof8Δ*, *thc1Δ*, and *bmc1Δ* strains. Genomic DNA was digested with EcoRI, separated by gel electrophoresis, transferred to a membrane, and probed for telomeric repeats and the *rad16⁺* locus as a loading control (LC).

biochemical experiments and the identifications of those RNAs in fission yeast that are modified by Bmc1 and can serve as positive controls for loss of methyltransferase activity.

**Pof8, Thc1, and Bmc1 function in the same pathway.** The differences in the extent to which TER1 levels and telomere length were reduced following the deletion of *pof8*, *thc1*, or *bmc1*, led us to investigate whether they have distinct roles during TER1 biogenesis. We constructed otherwise isogenic single, double, and triple mutants to compare TER1 steady-state levels by RT-qPCR and telomere length by Southern blotting. Neither TER1 levels (Fig. 4a) nor telomere length (Fig. 4b) were further reduced in any double or the triple deletion combination compared to the single

*pof8* deletion. It is noteworthy that telomeres in a *pof8* deletion are already critically short as the terminal restriction fragment is almost entirely comprised of subtelomeric DNA and chromosome end fusions are readily detected in a *pof8Δ* culture[17]. Any exacerbation of this phenotype would result in telomere loss and chromosome circularization as observed in *trt1Δ* cells[48].

**Deletion of *pof8* affects stability and interaction between Thc1 and Bmc1.** To further understand the difference in phenotype among the single gene deletions, we assessed the stability of the other subunits and integrity of the complex in the absence of each protein. Deletion of Pof8 resulted in an approximately fourfold reduction in Bmc1 (Fig. 4c) and no Bmc1 was detected in Thc1

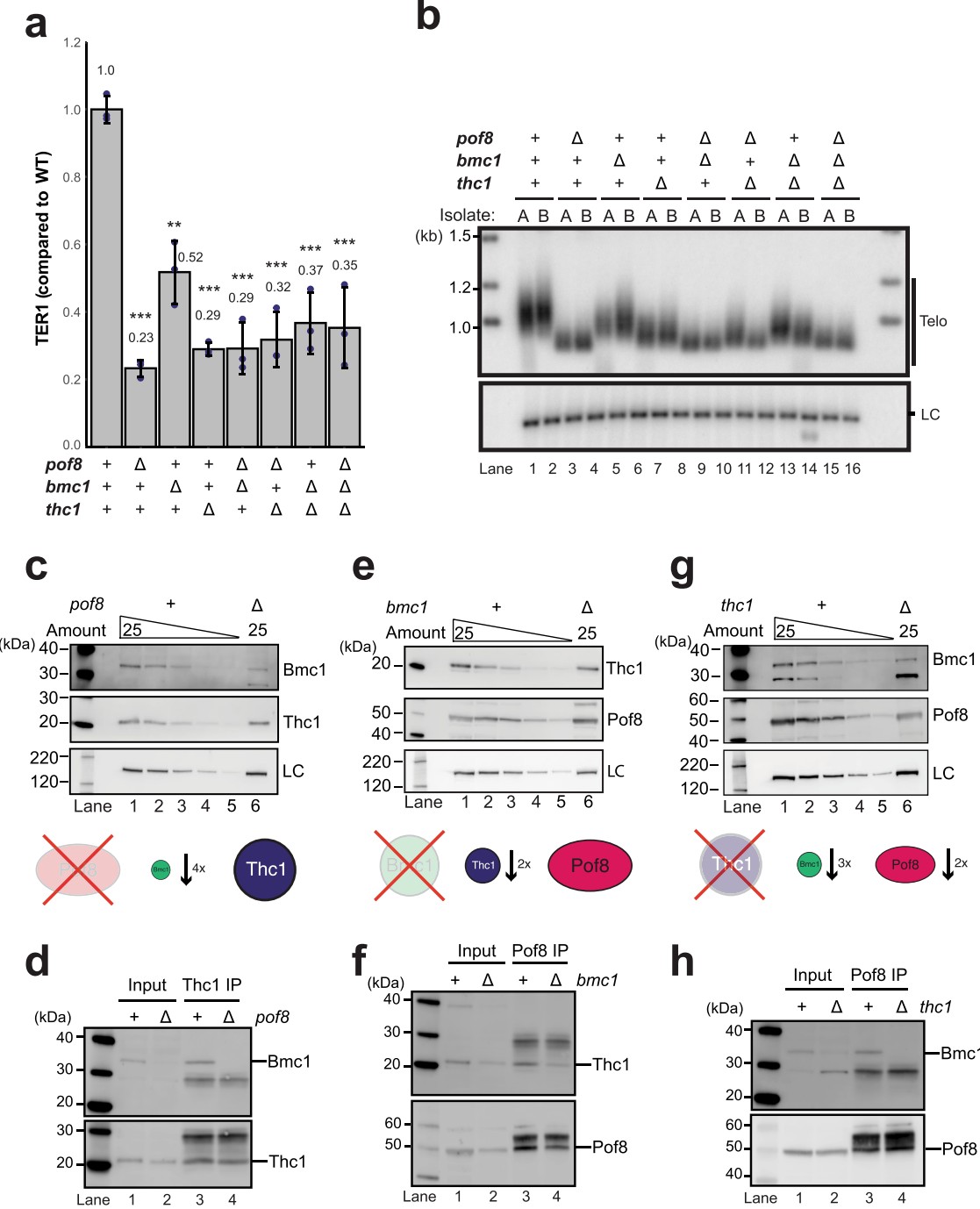

**Fig. 4 Functional interactions among Pof8, Thc1, and Bmc1. a** Effect of single, double, and triple deletions of *pof8*, *thc1*, and *bmc1* genes on TER1 levels measured by RT-qPCR. Mean (±SD, $n = 3$) fold-change values relative to wild type (WT). Significant differences from WT are marked above each bar (two-sided Dunnett's test; ***$p < 0.001$, **$0.001 \leq p < 0.01$, *$0.01 \leq p < 0.05$). **b** Telomeric Southern blot of the strains used in **a**. Two independent isolates of each deletion are shown. The *rad16*+ locus was used as a loading control (LC). **c** Pof8 stabilizes Bmc1. Western blot analysis for Bmc1-Twinstrep and Thc1-2xV5 from 25 μg of extract of a *pof8*Δ strain and a twofold serial dilution of extract from a *pof8*+ control strain using anti-strep-tag II and anti-V5 antibodies. An unknown protein recognized by the anti-strep-tag II antibody was used as a loading control (LC). **d** Pof8 is required for the Thc1–Bmc1 interaction. Western blot analysis of co-immunoprecipitation using anti-V5 in strains containing Bmc1-Twinstrep and Thc1-2xV5 in *pof8*+ and *pof8*Δ backgrounds. The blot was probed with anti-Strep-tag II antibody for Bmc1-Twinstrep (*top*) and reprobed with anti-V5 antibody for Thc1-2xV5 (*bottom*). Input represents 10% of IP. **e** Bmc1 stabilizes Thc1. Assessment of steady-state protein levels as in **c** for Thc1-2xV5 and 3xFLAG-Pof8 following the deletion of *bmc1*. **f** Bmc1 is not required for Thc1–Pof8 interaction. Western blot analysis of co-IP using anti-FLAG antibody in strains containing 3xFLAG-Pof8 and Thc1-2xV5 in *bmc1*+ or *bmc1*Δ backgrounds. The blot was probed with anti-V5 antibody for Thc1-2xV5 (top) and reprobed with anti-FLAG for 3xFLAG-Pof8 (bottom). **g** Thc1 stabilizes Bmc1 and Pof8 protein. Assessment of steady-state protein levels as in **c** for Bmc1-Twinstrep and 3xFLAG-Pof8 following the deletion of *thc1*. **h** Thc1 is required for the Pof8-Bmc1 interaction. Western blot analysis of co-IP using anti-FLAG in strains containing 3xFLAG-Pof8 and Bmc1-Twinstrep in *thc1*+ or *thc1*Δ backgrounds. The blot was probed with anti-Strep-tag II for Bmc1-Twinstrep (top) and reprobed with anti-FLAG for 3xFLAG-Pof8 (bottom).

immunoprecipitates (Fig. 4d, compare lanes 3 and 4). When protein extract was prepared from *bmc1Δ* cells, Thc1 levels were approximately twofold reduced, whereas Pof8 levels were close to wild type (Fig. 4e). Despite the reduction in Thc1, the protein still co-immunoprecipitated with Pof8 (Fig. 4f). In contrast, deletion of *thc1* resulted in a threefold decrease of Bmc1 protein and a twofold reduction in Pof8 (Fig. 4g). Although Bmc1 was readily detected in the input, no Bmc1 was observed in the Pof8 immunoprecipitate (Fig. 4h). These results support physical and functional interactions within the complex, in which the stability of individual factors and the integrity of the complex depend on the presence of the other members. Most importantly, the association of Bmc1 with the complex depends on the presence of Pof8 and Thc1. In the absence of either Pof8 or Thc1, the interaction of Bmc1 with the remaining factor is compromised. The quantitative differences in the effects on TER1 levels and telomere length between the single gene deletions can be explained by the loss of Pof8 or Thc1 causing the associated loss of Bmc1.

Further supporting the importance of stabilizing interactions between Thc1 and Pof8, recombinant 6xHis-Thc1 and untagged Pof8 co-expressed in *Escherichia coli* were readily purified as a complex (Fig. 5a). Whereas IP of the Lsm2-8 complex from extracts lacking endogenous Pof8 results in poor recovery of TER1 RNA and telomerase activity, the addition of purified Pof8-Thc1 complex to the cell extract was sufficient to increase the amount of TER1 recovered from an Lsm4-cMyc IP by fourfold and telomerase activity by up to ninefold (Fig. 5b).

**Effects on the Sm to Lsm2-8 switch**. Biochemical studies with purified proteins showed that Pof8 and Lsm2-8 interact directly and bind telomerase RNA cooperatively[32]. To assess to what extent Thc1 and Bmc1 affect the association of Pof8 and Lsm2-8 with TER1 in vivo, we immunoprecipitated the Lsm complex via c-Myc-tagged Lsm4 from extract prepared from cells deleted for *pof8*, *thc1* and *bmc1*. The deletions neither affected the level of Lsm4 protein in the extract nor the amount of immunoprecipitated Lsm4 (Fig. 5c). Nevertheless, the amounts of co-precipitating TER1 showed striking differences depending on the genetic background. Confirming previous results, deletion of *pof8* causes a 100-fold reduction in the amount of TER1 associated with the Lsm complex (Fig. 5d). In stark contrast, deletion of *thc1* only caused a twofold reduction in Lsm–TER1 association relative to the wild-type control. Deletion of *bmc1* even enhanced the interaction, albeit slightly. Interestingly, despite the modest effect on TER1-Lsm association, loss of Thc1 resulted in a 15-fold reduction in telomerase activity associated with Lsm4 (Fig. 5e). Under the same conditions, the activity was undetectable in the absence of Pof8 and was two to fourfold higher than wild type in the absence of Bmc1. These findings suggest that Thc1 contributes to the assembly of active telomerase by enhancing the ability of Pof8 to recognize correctly folded TER1 or by providing an additional level of quality control, perhaps by assessing the processing state at the 5' end of the RNA.

Prior to spliceosomal cleavage, the Sm complex binds the $AU_6GG$ motif, which after cleavage becomes the oligo-uridine binding site for Lsm2-8. To assess whether members of the Pof8-Thc1-Bmc1 complex also affect the stability of the Sm complex on TER1, we assessed the amount of RNA precursor that is associated with Smb1. Although recovery of TER1 varied by up to 1.5-fold among strains, none of these differences were statistically significant (Fig. 5f). We concluded that Sm complex association with the TER1 precursor is unaffected by Pof8, Thc1, and Bmc1 consistent with this complex functioning downstream of spliceosomal cleavage. Consistent with this notion, an association

of Smb1 with the first exon of TER1 (mature form) was increased in each of the three deletions (Fig. 5g). These observations support a model in which the transition from Sm-bound to Lsm-bound form is an active process and that TER1 remains associated with the Sm complex for longer in the absence of Pof8, Thc1 or Bmc1.

**Thc1 and Bmc1 stabilize Pof8 on telomerase and are associated with telomerase activity**. With the deletion of *pof8* affecting Lsm recruitment, telomerase activity, and telomere maintenance more strongly than *thc1Δ* and *bmc1Δ*, we next assessed the extent to which Thc1 and Bmc1 affect the association of Pof8 with TER1. Every single deletion caused a reduction in the amount of TER1 that co-immunoprecipitated with Pof8 (Fig. 6a). A 15-fold reduction was observed when both *thc1* and *bmc1* were absent, demonstrating that the stability of the Pof8-TER1 complex is clearly enhanced by the presence of Thc1 and Bmc1. Consistent with the co-IP data, telomerase activity associated with Pof8 was also reduced in the absence of either Thc1 or Bmc1 alone, or when both were absent (Fig. 6b). Although Lsm2-8 and Pof8 clearly still bind TER1 to some extent in the absence of Thc1 or Bmc1, the stability of the interaction between Pof8 and the core components of telomerase appears to be more transient in the absence of either protein.

To assess whether Thc1 and Bmc1 themselves are only transiently associated with the Lsm2-8 complex and Trt1, we precipitated Lsm4-cMyc from the extract in which either Thc1 or Bmc1 were also epitope-tagged. Thc1 and Bmc1 were readily detected in Lsm4 immunoprecipitates (Fig. 6c, d). Further supporting that Thc1 and Bmc1 are associated with active telomerase, we were able to detect telomerase activity following IP of epitope-tagged versions of Thc1 and Bmc1 (Supplementary Fig. 5). The activity associated with Thc1 was comparable to Pof8 IPs, while a reduced level of activity was recovered from Bmc1 IPs. However, this result must be interpreted with caution as different epitope tags are present on Pof8, Thc1, and Bmc1.

To gain insight into what fraction of telomerase is associated with Thc1 and Bmc1 respectively, we first asked whether either protein can be immunodepleted from the extract in a single step. Comparison of input and IP supernatant revealed that over 90% of Thc1 is removed from the soluble fraction in a single round of IP (Fig. 6e). Similarly, the supernatant is efficiently depleted of Bmc1 in a single round of IP (Fig. 6f). Using extracts from cells in which Thc1 and Lsm4 were both epitope tagged, we first immunoprecipitated Thc1 and then immunoprecipitated Lsm4-cMyc from the IP supernatant to assess how much telomerase activity remains in the supernatant following depletion of Thc1. To account for the non-specific reduction in telomerase activity associated with each round of IP, first-round mock IPs were carried out in parallel. Relative to mock depleted extract, 50% of telomerase activity was detected in Lsm4 IP after extracts had first been depleted of Thc1 (Fig. 6g). This is in stark contrast to the only 1% of activity detected when the extract was first depleted of Pof8[17]. Even more strikingly, the analogous experiment performed with Bmc1 revealed that close to 90% of telomerase activity was recovered with Lsm4 from the supernatant of a Bmc1 immunodepletion (Fig. 6g). These results indicate that, unlike Pof8 and Lsm2-8, Thc1, and even more so, Bmc1, are only loosely associated with active telomerase.

**Deletion of *thc1* affects the expression of several ncRNAs including TER1**. Considering the sequence similarity of Thc1 and Bmc1 with general RNA processing factors in other species, we asked whether the expression of RNAs other than TER1 is also affected by the deletion of either gene. For the deletion of *thc1*, 77

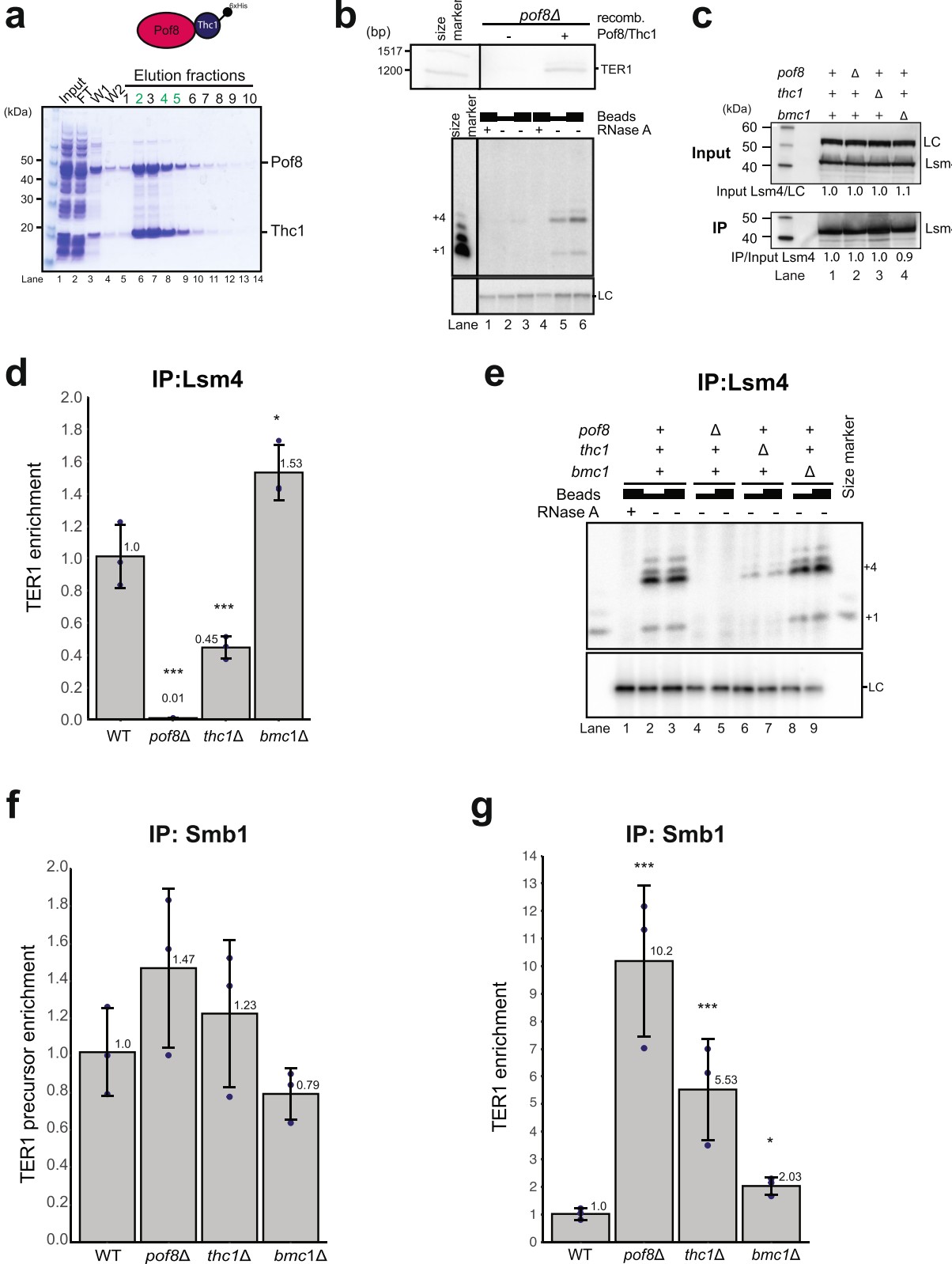

genes met the criteria of a twofold change in expression level with an associated adjusted *p* value of <0.05 (Supplementary Fig. 6a). Next to TER1 and *thc1*, 26 loci, including 11 non-coding RNAs were downregulated (Supplementary Table 2a). We also identified 49 loci that were upregulated, the majority of which were non-protein coding (Supplementary Table 2b). It therefore appears likely that Thc1 is involved in the biogenesis or regulation of

additional RNAs than just TER1. Far fewer changes in gene expression were observed following the deletion of *bmc1* with no loci meeting the criteria described above (Supplementary Fig. 6b). In light of U6 snRNA being a target of vertebrate MePCE, we specifically examined this locus but found no significant change in expression in either deletion. However, it must be noted that the knockdown of MePCE in human cells also did not reduce the

**Fig. 5 Requirements for Lsm and Sm loading onto TER1. a** Coomassie-stained SDS-PAGE gel of immobilized-metal affinity chromatography (IMAC) purified recombinant 6xHis-Thc1 and untagged Pof8 co-expressed from a multicistronic plasmid in *E. coli*. The gel shows the input (lane 1), flow-through (FT, lane 2), two washes (W, lanes 3 and 4), and 10 fractions of eluate (lanes 5–14). Fractions in green were pooled, quantified, and used in reconstitution experiments. **b** Extracts from *pof8Δ* cells were mixed with buffer or recombinant Pof8/6xHis-Thc1 complex. Northern blot of TER1 following immunoprecipitation of Lsm4-cMyc (top). Telomerase activity assay with same IP fractions (bottom). RNase A was added in lanes 1 and 4 as controls; loading control (LC): $^{32}$P-labeled 100-mer oligonucleotide **c** Steady-state level of Lsm4 is not affected by *pof8Δ*, *thc1Δ*, or *bmc1Δ*. Extracts from cells harboring the deletions were analyzed by western blot with anti-cMyc (top); LC: α-tubulin. Similar amounts of Lsm4-cMyc were precipitated from all strains (bottom). **d** Pof8 and Thc1—but not Bmc1—are required for loading the Lsm2-8 complex onto TER1. RT-qPCR of TER1 from immunoprecipitation of Lsm4-cMyc from extracts of wild type (WT), *pof8Δ*, *thc1Δ*, and *bmc1Δ* cells. Bars represent mean (±SD, $n = 3$) fold-enrichment relative to WT after normalization to the input levels. Significant values are marked above each bar (Two-sided Dunnett's test; ***$p < 0.001$, **$0.001 \leq p < 0.01$, *$0.01 \leq p < 0.05$). **e** Telomerase activity assay using the same Lsm4 IP samples as in **d** shows activity is compromised in the absence of Pof8 or Thc1. Ten and 20 μl of IP suspensions were used; 20 μl of IP suspension was used for RNase A treatments. **f** Association of Sm proteins with TER1 precursor is not affected in the absence of Pof8, Thc1, or Bmc1. RT-qPCR of TER1 precursor following IP of Smb1-cMyc from WT, *pof8Δ*, *thc1Δ*, and *bmc1Δ* cells. Bars represent mean (±SD, $n = 3$) fold-enrichment relative to WT after normalization to input levels. **g** The Sm complex association with TER1 following spliceosomal cleavage is stabilized to varying degrees in the absence of Pof8, Thc1, or Bmc1. RT-qPCR of TER1 following IP of Smb1-cMyc from WT, *pof8Δ*, *thc1Δ*, and *bmc1Δ* cells. Bars represent mean (±SD, $n = 3$) fold enrichment relative to WT after normalization to input levels. Statistical analysis as in **d**.

steady-state levels of U6 snRNA[39]. While changes in expression levels are indicative of a specific RNA being a target for modification, RNA stability is not necessarily affected by the absence of the methyl group. It therefore will require further studies to assess whether Bmc1 functions specifically as a telomerase biogenesis factor or has additional functions in fission yeast.

## Discussion

Previous work demonstrated that the *S. pombe* Pof8 protein is a LARP7 family member that functions in telomerase biogenesis by facilitating the loading of the Lsm2-8 complex onto TER1 through cooperative protein–protein and protein–RNA interactions[15–17,32]. We now reveal that Pof8 carries out this function in complex with Thc1 and Bmc1, two previously uncharacterized proteins. Deletion of *thc1* or *bmc1* results in a decrease in Pof8 association with telomerase, decreased levels of TER1, and shorter telomeres. Although physical interactions support that Pof8, Thc1, and Bmc1 function together during telomerase assembly, they clearly have distinct roles in biogenesis with Pof8 remaining associated more tightly with active telomerase than Thc1 and Bmc1. It is further worth noting that the levels of telomerase activity associated with Pof8 in the extract from *bmc1Δ* cells were significantly lower compared to cell extracts from *bmc1+* cells (Fig. 6b). In contrast, the Lsm4 associated telomerase activity was slightly increased in the absence of Bmc1, suggesting that Bmc1 stabilizes the binding of Pof8 to telomerase but is not required for the stable association of the Lsm complex (Fig. 5e). These results support a model where the interaction of Pof8 with TER1 is stabilized by Thc1 and Bmc1, and both factors contribute to the assembly and stability of functional telomerase but do not necessarily remain associated with the active enzyme (Fig. 7).

To our knowledge, this is the first example of a member of the LARP7 family functioning in a complex with MePCE/Bin3 outside the 7SK complex, indicating that the functional association of these two proteins predates the metazoan invention of the 7SK RNP as a transcriptional and translational regulator. Coincidentally, RIP-Seq experiments with epitope-tagged Bmc1 (Bin3) also identified TER1 as a target leading to an independent characterization of the Bmc1/Pof8 interaction by the Bayfield group[42].

It has been widely assumed that the 7SK complex emerged in metazoans to regulate transcription elongation, a process that gained in importance as intron size, and thereby primary transcript length, increased. This assertion is in part based on the inability to locate 7SK non-coding RNAs in the genomes of yeast and other unicellular organisms, but the high degree of divergence between vertebrate 7SK RNAs and those found in arthropods[49] justifies the old adage that absence of evidence is not evidence of absence. In light of well over 100 non-coding RNAs of unknown function in the fission yeast genome, it remains to be seen whether a 7SK-like complex exists here and in other unicellular organisms.

Should this not be the case, one will have to look elsewhere for the evolutionary origins of Larp7 and MePCE/Bin3 proteins functioning together in non-coding RNA processing or as stable components of a functional RNP. Prior to the discovery of a MePCE/Bin3 family member functioning in telomerase biogenesis, the presence of Larp7 family members as constitutive components of telomerase holoenzymes in evolutionarily distant unicellular species already suggested that the association of Larp7 with this RNP predates the existence of a 7SK complex. With Bmc1 and Pof8 functioning together in telomerase biogenesis, it is clear that the Larp7–MePCE/Bin3 partnership has ancient roots. Interestingly, the stability of Bmc1 is largely dependent on the presence of Thc1 and Pof8 indicating that most if not all Bmc1 in the cell is in complex with these two proteins and they are likely forming a functional unit on whichever RNAs they associate with.

It was recently reported that the 7SK complex interacts with the SMN complex in the cytoplasm to regulate the assembly of Sm complexes on snRNAs to connect spliceosome production with transcriptional output[50]. It is thus noteworthy that telomerase RNAs in yeast share features with snRNAs, including the presence of Sm-binding sites and TMG cap structures[6,7,51]. Whether this resemblance was critical in telomerase acquiring these biogenesis factors or whether telomerase is the more ancient target RNA remains to be investigated.

LARP7 family members share an atypical RNA recognition motif, named xRRM for the presence of an extended α3 helix at the C-terminus that lies across the β-sheet[52]. While the C-terminal RRM in Pof8 shares most features with the xRRM domains in p65 and human Larp7, such as the presence of the conserved RNP3 sequence Y/W-X-D/Q on β2, a conserved arginine on β3, and charged/aromatic residues on the C-terminal end of helix α3, the C-terminal helix itself is considerably shorter in Pof8 than in other family members. This may result in reduced RNA binding affinity as the extended α3 helix in Larp7 and p65 create a conserved binding pocket between the third and fourth turns of the α3-helix. Truncation of the α3-helix results in reduced binding affinity to substrate RNA[53], leading to the suggestion that the shorter α3-helix in Pof8 will also result in a weaker binding affinity for RNA than seen with p65 and hLarp7[31]. The α3 helix in Pof8 is nevertheless important for RNA binding as mutations lower the affinity for TER1, reduce TER1

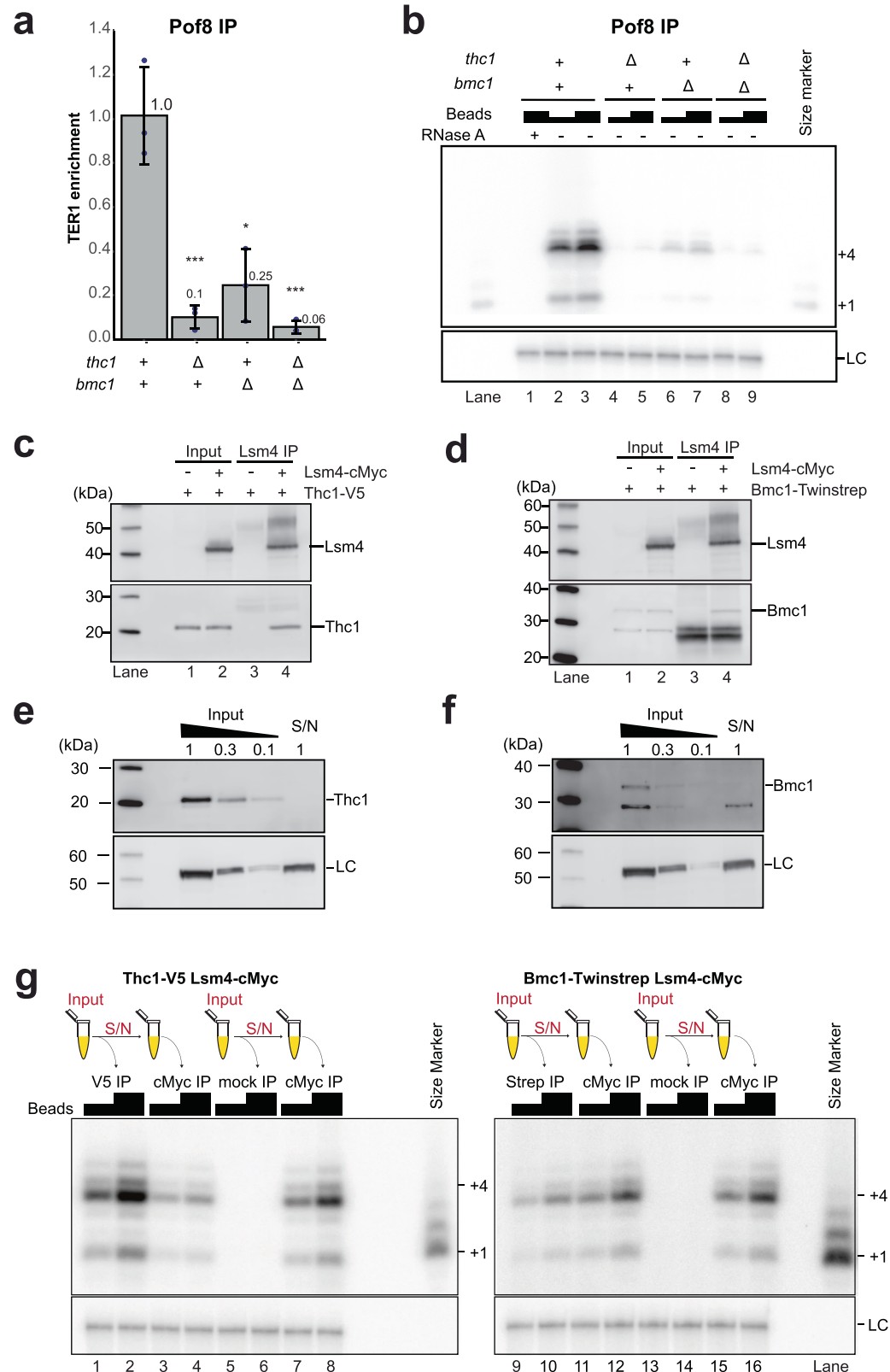

levels and cause telomere shortening similar to a deletion of the entire open reading frame[15,16,32]. Recent structural and biochemical analysis revealed that TER1 interactions with RRM1 and xRRM2 of Pof8, as well as protein interactions with the Lsm2-8 complex, function cooperatively in binding to TER1[32]. However,

the addition of recombinant Pof8 to extracts from *pof8Δ* cells was insufficient to restore telomerase activity. In contrast, we find that the addition of a preformed Pof8-Thc1 complex increased the association of TER1 with Lsm2-8 and strongly enhances in vitro telomerase activity (Fig. 5b). Together with our finding that *thc1*

**Fig. 6 Thc1 and Bmc1 stabilize Pof8 on telomerase and are associated with Lsm proteins and active telomerase. a** The absence of Thc1 and/or Bmc1 weakens the Pof8–TER1 interaction. RT-qPCR of TER1 following IP of FLAG-tagged Pof8 from *thc1Δ*, *bmc1Δ*, and combined deletion backgrounds relative to wild-type (WT) background. The bars represent the mean fold enrichment of TER1 relative to WT after normalization to TER1 levels in the input. Error bars represent standard deviation based on three biological replicates. **b** Telomerase activity assay from Pof8 IP samples of the same strains used in **a**. IP suspensions (10 and 20 μl) were used for telomerase activity assays; 20 μl of IP suspension was used for the RNase A treatment. **c** Thc1 co-immunoprecipitates with Lsm4. Extracts from strains containing V5 epitope-tagged Thc1 and untagged or cMyc-tagged Lsm4 were subjected to immunoprecipitation with anti-c-Myc followed by western blotting. The blot was probed with anti-V5 and reprobed with anti-cMyc antibodies. **d** Bmc1 co-immunoprecipitates with Lsm4. Extracts from strains containing TwinStrep tagged Bmc1 and untagged or Lsm4-cMyc were immunoprecipitated with anti-c-Myc followed by western blotting. The blot was probed with anti-strep-tag II and reprobed with anti-cMyc. **e** Depletion of Thc1-2xV5 from cell-free extract in a single round of immunoprecipitation with anti-V5 on Dynabeads; 20 μg of IP supernatant (S/N) was loaded next to a serial dilution of input to assess the extent of depletion; loading control (LC): α-tubulin **f** Depletion of Bmc1-Twinstrep from cell-free extract in a single round of immunoprecipitation with anti-Strep-tag II on Dynabeads analyzed as in **e**. **g** Telomerase activity assay following sequential immunoprecipitations to assess the fraction of telomerase activity associated with Thc1 and Bmc1. The extract containing Thc1-2xV5 and Lsm4-cMyc was immunoprecipitated with anti-V5 (V5 IP) or anti-FLAG (mock IP). The IP S/N was then immunoprecipitated with anti-cMyc (cMyc IP). IP suspensions (10 and 20 μl) were used for telomerase activity assays. The extract containing Bmc1-Twinstrep and Lsm4-cMyc was immunoprecipitated with anti-strep-tag II (strep IP) or anti-FLAG (mock IP). The IP S/N was then immunoprecipitated with anti-cMyc (cMyc IP) and analyzed as described above.

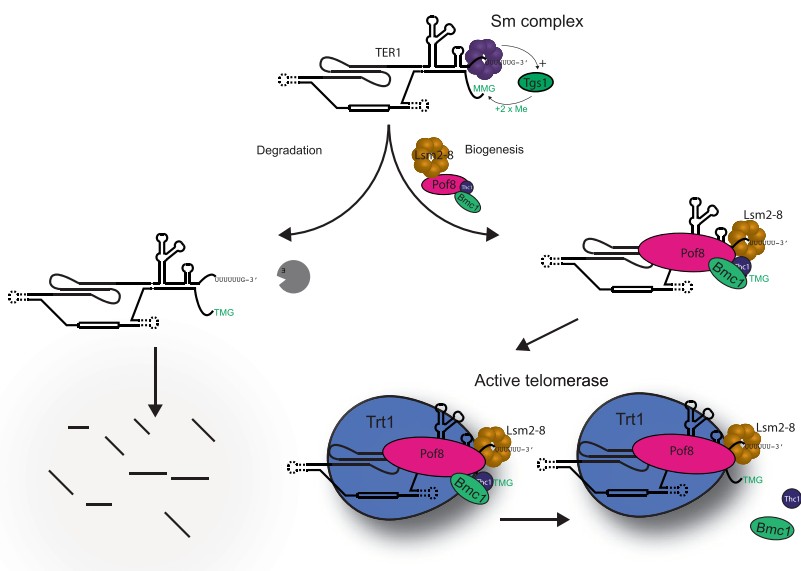

**Fig. 7 Proposed model of the role of Thc1 and Bmc1 during the stepwise assembly of telomerase.** After spliceosomal cleavage, the Sm complex (purple) facilitates cap hypermethylation by Tgs1 and is subsequently replaced by the Lsm2-8 complex (brown). Pof8, Thc1, and Bmc1 function during the Sm to Lsm transition. Pof8 (pink) promotes the loading of Lsm2-8 onto TER1, while Bmc1 (green) and Thc1 (blue) stabilize Pof8 binding through protein–protein and likely through protein–RNA interactions. In the absence of Pof8 on TER1, the Lsm2-8 complex cannot load, leading to TER1 degradation. Whereas Pof8 and Lsm proteins are quantitatively associated with active telomerase, Thc1 and Bmc1 are only associated with a fraction of active enzyme.

deletion results in a 10-fold decrease in Pof8-associated TER1 (Fig. 6a), these results suggest that Thc1 and Pof8 bind the RNA cooperatively.

Finally, the similarity of Thc1 with the cap-binding domain of NCBP3 and PARN raises the exciting possibility that Thc1 interacts with the 5' end of TER1, while Pof8 binds the pseudoknot region and Lsm2-8 the 3' end of TER1, thereby integrating information on the processing and folding status from distant regions of the RNA and providing quality control for subsequent biogenesis events such as the loading of the catalytic subunit. Together with recent studies tying LARP7-mediated U6 snRNA modification to splicing robustness and fidelity[23,24], the results presented here underscore the versatility of LARP-family proteins to function as molecular connectors in RNA processing and RNP assembly.

## Methods

**Strains and constructs**. *S. pombe* strains used in this study are listed in Supplementary Table 3. The *bmc1* deletion was generated by replacing the entire open reading frame with the NatMX6 resistance cassette in a diploid strain generated fresh by crossing PP137 and PP139. The knockout fragment was generated by fusion PCR using primers listed in Supplementary Table 4 and contained the NatMX6 cassette and ~750 base pair (bp) of homology upstream and downstream of the *bmc1* open reading frame. Cells were transformed as described[54] and allowed to recover in a shaker at 25 °C overnight. Diploid transformants were then selected on yeast extract low adenine (YEA) plates supplemented with nourseothricin (NAT, 100 μg ml[−1]). The correct insertion of the NatMX6 cassette was confirmed by PCR and haploid cells were obtained by sporulation and selection on YEA plus NAT plates. The absence of the *bmc1* ORF in the haploid cells was confirmed by PCR and sequencing.

The *thc1* deletion was generated by replacing the open reading frame with the *his3+* gene using primers listed in Supplementary Table 4 and the same strategy as above except that haploid strain PP138 was used for transformation. Epitope tags were introduced as follows: 3xFLAG tag was introduced at the N-terminus of Pof8,

2xV5 at the C-terminus of Thc1, and Twinstrep tag at the C-terminus of Bmc1. Additional strains were generated by crossing and selection of correct genotypes followed by verification by PCR or western blotting. A DNA fragment encompassing Twinstrep-tagged Bmc1 from position −408 to +682 was generated by PCR amplifying the genomic DNA derived from a Bmc1-Twinstrep tagged strain and cloned into pDBlet plasmid[55]. Point mutants were generated by site-directed mutagenesis in this context and were verified by Sanger sequencing. Plasmids were introduced into *S. pombe* cells by electroporation.

**Protein extracts**. Cultures (2 L) were grown to a density of 0.5–1 × $10^7$ cells per ml and harvested by centrifugation. Cells were washed three times with ice-cold TMG(300) buffer (10 mM Tris-HCl pH 8.0, 1 mM magnesium chloride, 10% (v/v) glycerol, 300 mM sodium acetate), resuspended in two packed cell volumes of TMG(300) plus supplements (complete EDTA-free protease inhibitor cocktail (Roche), 0.5 mM PMSF, 1 mM EDTA, and 0.1 mM dithiothreitol (DTT)), and quick-frozen by dripping the cell suspension into liquid nitrogen. Cells were lysed in a 6850 Freezer mill (SPEX SamplePrep) using eight cycles (2 min) at a rate of 10 per second with 2 min cooling time between cycles. Lysates were thawed on ice and one additional packed cell volume of TMG(300) plus supplements was added. Lysates were then cleared by centrifugation twice for 10 min at 6000 × *g* in a Sorvall Linx 6000 with F13-14x50cy rotor and then once for 50 min in a Beckman 70Ti rotor at 36,000 × *g*. All steps were carried out at 4 °C. For the affinity purification followed by mass spectrometry experiment the extract was prepared as above with the following modification: the lysis buffer contained 50 mM Tris-HCl pH 7.5, 150 mM sodium chloride, 0.1% (v/v) NP40 and 10% (v/v) glycerol, complete EDTA-free protease inhibitor cocktail (Roche), 0.5 mM PMSF, 1 mM EDTA, and 0.1 mM DTT. Cell lysates were centrifuged for 15 min at 8000 × *g* in a Beckman JA-17 rotor immediately after thawing. The supernatant was collected, and one additional packed cell volume of lysis buffer was added to the pellet and centrifuged for 15 min at 12,000 × *g* in a Beckman JA-17 rotor. The two supernatants were pooled together and then cleared by centrifugation for 40 min in a Beckman 70Ti rotor at 40,000 × *g*. For all extracts, protein concentrations were determined by Bradford assay and ranged from 5 to 10 mg ml⁻¹.

Denatured protein extracts for western blotting were prepared from 1 × $10^8$ cells through lysis by vortexing with 0.5-mm glass beads in 10% trichloroacetic acid for 8 min at 4 °C. Beads were washed once with 0.8 ml 10% trichloroacetic acid, the bead supernatant was transferred to a fresh tube, and protein precipitate was collected by centrifugation at 16,000 × *g* for 2 min. Protein pellets were washed once with acetone and resuspended in 120 µl of 1× protein sample buffer (1× NuPAGE LDS buffer, 50 mM DTT, 2% [w/v] sodium dodecyl sulfate). Samples were incubated for 5 min at 75 °C and centrifuged at 16,000 × *g* for 1 min. The soluble fraction was collected and analyzed by sodium dodecyl sulfate (SDS)-polyacrylamide gel electrophoresis (PAGE) and western blotting.

**Immunoprecipitations**. For the identification of proteins that interact with Pof8, cell-free extracts of strains FP1547 harboring 3xFLAG epitope-tagged Pof8 and untagged control (FP1546) were diluted to 5 mg ml⁻¹ in 8 ml of lysis buffer (50 mM Tris-HCl pH 7.5, 150 mM sodium chloride, 0.1% [v/v] NP40 and 10% [v/v] glycerol) plus complete EDTA-free protease inhibitor cocktail (Roche), 0.5 mM PMSF, 1 mM EDTA, and 0.1 mM DTT and incubated with 500 µl of EZview Red Flag-agarose 50% slurry equilibrated in lysis buffer (Sigma, F2426). The mix was incubated for 8 h at 4 °C with gentle rotation. Agarose beads were collected by centrifugation at 300 × *g* for 5 min with low brake and washed 5 times for 5 min with 5 ml of wash buffer (50 mM Tris-HCl pH 7.5, 250 mM sodium chloride, 0.1% [v/v] NP40, and 10% [v/v] glycerol) at 4 °C with gentle rotation. Proteins were eluted from beads by incubation for 30 min with 500 µl elution buffer (50 mM Tris-HCl pH 7.5, 250 mM sodium chloride, 0.1% [v/v] NP40 and 10% [v/v] glycerol, and 100 µg ml⁻¹ 3× FLAG peptide) at 4 °C. The elution step was repeated two more times and the eluates were pooled. A 4 µl aliquot of the first eluate was used for Silver stained SDS-PAGE analysis following the manufacturer's instructions (SilverXpress Silver Staining Kit; Invitrogen, LC6100).

**Protein digestion**. The immunoprecipitated protein eluates were treated with 0.1 U of benzonase (Sigma, E8263) for 30 min at 37 °C and further diluted with the same volume of 100 mM Tris-HCl, pH 8.5. The samples were then split into three 400 µl aliquots, mixed with 100 µl of 100% trichloroacetic acid (TCA), and incubated at 4 °C overnight. After centrifugation at 20,000 × *g* for 30 min at 4 °C, the pellets were washed twice with 500 µl cold acetone and centrifuged at 20,000 × *g* for 10 min after each wash. Air-dried pellets were resuspended in 30 µl of 100 mM Tris-HCl, pH 8.5, 8 M Urea, and pooled. 4.5 µl of 0.1 M Tris(2-carboxyethyl)-phosphine hydrochloride (TCEP) was added to the pooled solution to a final concentration of 5 mM and incubated for 30 min at room temperature (rt). After the incubation, 1.8 µl of 0.5 M CAM was added and incubated for 30 min at rt in the dark. Proteins were then digested with 1 µl of 0.1 µg µl⁻¹ Lys-C at 37 °C overnight. The samples were then diluted to 2 M Urea with 100 mM Tris-HCl, pH 8.5, added with CaCl₂ to 2 mM and digested with 5 µl of 0.1 µg µl⁻¹ Trypsin at 37 °C overnight. Finally, 90% formic acid was added to a final concentration of 5% and samples were analyzed by MudPIT mass spectrometry as described[56].

**Liquid chromatography-tandem mass spectrometry**. Peptides were loaded onto a 100-µm fused silica (Polymicro Technologies) capillary column packed with 3 cm of 5-µm reverse-phase C18 resin (Aqua; Phenomenex), 4 cm of 5-µm strong cation exchange resin (Luna; Phenomenex), and 8 cm of reverse-phase C18 resin[57]. Loaded microcapillaries were placed in line with an Agilent 1100 quaternary pump (Agilent Technologies, Palo Alto, CA). Overflow 50-µm fused silica tubing was used to decrease the flow rate from 0.1 ml min⁻¹ to about 200–300 nl min⁻¹. Three different elution buffers were used: 5% acetonitrile, 0.1% formic acid (Buffer A); 80% acetonitrile, 0.1% formic acid (Buffer B); and 0.5 M ammonium acetate, 5% acetonitrile, 0.1% formic acid (Buffer C). Eleven cycles with increasing salt concentrations followed by organic gradients slowly released peptides directly into the mass spectrometer[56]. The first chromatography step only implemented the reversed-phase gradient starting with a 10 min 100% A wash; from 0 to 40% B between 10 and 80 min; to 100% B from 80 to 90 min; to 100% A from 90 to 100 min. Steps 2–8 used a short salt bump of increasing buffer C concentration set at *X* = 15, 20, 30, 40, 60, 70, and 80%; starting with a 3-min 100% A wash; *X*% C and 100 − *X*% A from 3 to 5 min; to 100% A from 5 to 10 min; to 15% B from 10 to 25 min; to 45% B from 25 to 117 min. Chromatography steps 9 and 10 comprised a long high salt wash starting with a 2-min 100% A wash; 100% Buffer C from 2 to 22 min; followed by a 5-min 100% A wash; followed by the acetonitrile gradient to 20% B from 27 to 37 min; to 70% B from 37 to 105 min; to 100% B from 105 to 115 min; back to 100% A from 115 to 117 min. The last step was a 2-h wash with buffer A.

The application of a 2.5 kV distal voltage electro-sprayed the eluting peptides directly into an LTQ linear ion trap mass spectrometer (Thermo, San Jose, CA) equipped with a custom-made nano-LC electrospray ionization source (Florens and Washburn 2006). Full MS spectra were recorded over a 400 to 1600 *m/z* range, followed by five tandem mass (MS/MS) events sequentially generated in a data-dependent manner on the first to fifth most intense ions selected from the full MS spectrum (at 35% normalized collision energy). Data were acquired in positive ion and centroid mode with activation Q energy set at 0.250, activation time of 30 ms, and an isolation width of 1 *m/z*. Dynamic exclusion was enabled for 120 s with a repeat count of 2 and an exclusion list size of 500.

RAW files were extracted to peak lists in.ms2 file format[58] using RawDistiller v.1.0[59]. MS/MS spectra were searched using ProLuCID v. 1.3.3[60] with a mass accuracy of 500 ppm for both precursor and fragment ions. Trypsin specificity was imposed on both ends of candidate peptides with up to 3 missed cleavages. Static modification was imposed for cysteine alkylation (+57 Da) and variable modification was allowed on methionine for oxidation (+16 Da). Tandem mass spectra were searched with these parameters against a protein sequence database containing 5084 *S. pombe* proteins downloaded from NCBI on 2017-03-07 and 193 sequences from common contaminants (human keratins, IgGs, proteolytic enzymes). To estimate false discovery rates (FDR), each protein sequence was randomized (keeping the same amino acid composition and length) and the resulting "shuffled" sequences were added to the database, for a total search space of 10554 amino acid sequences.

DTASelect and CONTRAST v. 1.9[61] were used to select and sort peptide–spectrum matches and to compare results from duplicate analyses of immunoprecipitated samples and 2 negative controls. Peptide/spectrum matches were retained only if they had a Delta CN score (DeltCN) of at least 0.8 and a minimum cross-correlation score of 1.8 for singly, 2.2 for doubly, and 2.6 for triply charged spectra. Additionally, the peptides had to be a minimum of seven amino acids in length, fully tryptic, and mapped to at least 2 MS/MS spectra. With these criteria, the FDRs were 0.7 ± 0.7, 1.1 ± 0.2, and 3.8 ± 1.2 at the spectral, peptide, and protein levels, respectively. Proteins that were subsets of others were removed using the parsimony option in DTASelect on the proteins detected after merging all runs. Proteins that were identified by the same set of peptides were grouped together, and one accession number was arbitrarily considered as representative of each protein group. An in-house software, NSAF7 v. 0.0.1 (https://github.com/tzw-wen/kite/tree/master/windowsapp/NSAF7x64), was used to create the final reports on all detected peptides and nonredundant proteins identified across the different runs and calculate label-free distributed normalized spectral abundance factor (dNSAF) quantitative values for all detected protein/ protein groups[62].

**Co-immunoprecipitation**. For other IPs, cell-free extracts (5.5 mg) were diluted to 5 mg ml⁻¹ with TMG(300) buffer plus supplements. An aliquot (100 µl) was frozen as input control. Magnetic protein G Dynabeads (30 mg ml⁻¹; Invitrogen, 10004D) were coated with anti-c-Myc 9E10 (Sigma-Aldrich, M4439), anti-FLAG M2 (Sigma-Aldrich, F3165), anti-V5 (Thermo fisher, R960-25), or anti-strep-tag II (Abcam, ab76949) (20 µg per 100 µl of bead suspension) by incubation for 30 min at room temperature in 200 µl of 1× PBS + 0.1% (v/v) Tween-20. The magnetic beads and StrepTactin Sepharose (GE Healthcare, 28935599) were washed three times with 1 ml of TMG(300). IPs were performed with 60 or 75 µl (sequential IP) of bead suspension for Pof8, Thc1, and Bmc1 and 120 µl for Sm and Lsm for 4 h at 4 °C with gentle rotation. Recombinant Pof8/6xHis-Thc1 was added to 100 nM during the incubation for experiments in Fig. 5b. Beads were collected using a magnet or centrifugation at 300 × *g* for 2 min (StrepTactin Sepharose). The beads were then washed four times with 1 ml TMG(300) plus supplements and 0.1% (v/v) Tween-20, and once with TMG(50) plus supplements. Finally, beads were resuspended in 100 µl TMG(50) plus supplements and 0.4 U µl⁻¹ RNasin (Promega) and frozen in liquid nitrogen.

For co-IP analysis, the input aliquot (100 µl) was mixed with an equal volume of 2× protein sample buffer (2× NuPAGE LDS buffer [Life Technologies], 100 mM DTT, 4% [w/v] SDS) and the IP with 100 µl 1× protein sample buffer followed by incubation for 10 min at 75 °C. To assess whether the interaction is nucleic acid-dependent, the bead suspension was divided into 4 equal volumes, three samples were incubated in 20 µl of TMG(50), or plus 40 ng of RNase A (Invitrogen), or 500 U Benzonase (Sigma), respectively for 15 min at 37 °C and then washed twice with TMG(300) and TMG(50). All samples were resuspended in 25 µl of 1× protein sample buffer, incubated for 10 min at 75 °C, and supernatants were analyzed by western blotting.

**Western blot analysis**. Denatured protein extracts or immunoprecipitates were loaded onto 4–12% NuPAGE Bis-Tris gel (Life Technologies, NP0321BOX) for electrophoresis in 1× MOPS buffer (Life technologies, NP0001) at 160 V for 60 min. Proteins were transferred to Protran nitrocellulose membranes (Whatman) in western transfer buffer (3.03 g l$^{-1}$ Tris base, 14.4 g l$^{-1}$ glycine, 20% [v/v] methanol) at 100 V for 1 h with a cold pad stored at −20 °C prior to use. Blots were processed using an iBind Flex Western Device (Life Technologies, SLF20002). Primary antibodies were mouse anti-FLAG M2 (Sigma Aldrich, F3165) at 1:4000 dilution, rabbit anti-Strep-tag II (Abcam, ab76949) at 1:2000 dilution, rabbit anti-cMyc (Santa Cruz Biotechnology, sc-789) at 1:1000, and mouse anti-α-tubulin (Sigma, T5168) at 1:10,000 dilution. Primary antibodies for V5 epitope tag were rabbit anti-V5 (Abcam, ab9116) at 1:2000 dilution (Figs. 2 and 4d, f), mouse anti-V5 (Thermo Fisher, R960-25) at 1:5000 dilution (Figs. 4c, e and 6e), or mouse anti-V5-HRP (Thermo fisher, R961-25) at 1:1000 dilution (Fig. 6c). Secondary antibodies are horse-radish peroxidase-conjugated goat anti-mouse IgG (H + L) at 1:5000 (Thermo Scientific, 31430) and horse-radish peroxidase-conjugated goat anti-rabbit IgG (H+L) at 1:5000 (Thermo Scientific, 31460). Secondary antibodies were visualized with ECL plus substrate (Pierce, 32132) on a Typhoon FLA9500 (GE Healthcare Life Sciences) for co-IP blots and a ChemiDoc MP Imager (Bio-rad) for denatured protein extract blots. Image Lab (6.0.1) was used to quantify the western blots.

**RNA extraction**. For total RNA extraction, cell cultures (250 ml) were grown to a density of 6–8.5 × 10$^6$ cells ml$^{-1}$ and collected by centrifugation, washed twice with H$_2$O (500 ml), resuspended in 3 ml H$_2$O and quick-frozen by dripping the cell suspension into liquid nitrogen. Cells were lysed in a 6850 Freezer mill (SPEX SamplePrep) using seven cycles (2 min) at a rate of 10 per second with 2 min cooling time between cycles. Lysed cells were transferred into 50 ml tubes containing 5 ml phenol:chloroform:isoamyl alcohol (25:24:1, equilibrated with 50 mM sodium acetate, pH 5.2) and 5 ml 50 mM sodium acetate and 1% (w/v) SDS pre-heated to 65 °C. RNA was extracted four times with 5 ml phenol: chloroform:isoamyl alcohol (25:24:1, equilibrated with 50 mM sodium acetate, pH 5.2) and once with chloroform:isoamyl alcohol (24:1, equilibrated with 50 mM sodium acetate, pH 5.2). RNA was ethanol precipitated and resuspended in H$_2$O. In the context of IP experiments, RNA was isolated from input and beads by incubation with proteinase K (2 µg µl$^{-1}$ in 0.5% [w/v] SDS, 10 mM EDTA pH 8.3, 20 mM Tris-HCl pH 7.5) at 50 °C for 15 min, followed by extraction with phenol: chloroform:isoamyl alcohol and chloroform: isoamyl alcohol. RNA was ethanol precipitated for 4 h or overnight at −20 °C and dissolved in H$_2$O.

RNA used for RT-PCR was treated with 8 U of DNase I (RNase-free, NEB, M0303) for 15 min at 37 °C in a 50 µl reaction. The reaction volume was increased to 100 µl with H$_2$O and further purified using RNA Clean & Concentrator (Zymo Research, R1013) following the manufacturer's instructions. In brief, 100 µl RNA samples were mixed with 200 µl of RNA Binding Buffer and 300 µl of 100% ethanol. The mix was transferred to the Zymo-Spin IC Column and centrifuged for 30 s at 14,000 × g. The column then was washed with 400 µl of RNA Prep Buffer, and once with 700 µl of RNA Wash Buffer and centrifuge for 30 s at 14,000 × g followed by a final wash used 400 µl RNA Wash Buffer and centrifugation for 1 min at 14,000 × g. The flow-through was discarded in each step. The column was transferred to a new tube, and RNA was eluted in 20 µl of H$_2$O by centrifugation for 30 s at 14,000 × g.

**Northern blot analysis**. DNase-treated RNA isolated from IP was resuspended in 1× formamide loading buffer, incubated for 2 min at 75 °C, and separated on a 4% (v/v) polyacrylamide (29:1) gel containing 8 M urea and transferred to Biodyne nylon membrane (Pall Corporation) at 400 mA for 1 h in 0.5× TBE buffer. RNA was crosslinked to the membrane using 254-nm UV light at 120 mJ/cm$^2$ in a Stratalinker (Stratagene). Hybridization with radiolabeled probes (10 million counts per minute) were performed in Church–Gilbert buffer at 60 °C with TER1 probe (nucleotides 536–998, labeled with High Prime (Roche) and [α-$^{32}$P]-dCTP). Blots were washed briefly once with 100 ml of 0.1× SSC, 0.1% (w/v) SDS and twice with 100 ml of 0.1 SSC, 0.1% (w/v) SDS for 15 min at 60 °C each. Blots were exposed to PhosphorImager screens and analyzed with a Typhoon 8600 scanner. ImageQuant TL (v7.0) was used to quantify the northern blot.

**RT-qPCR**. Purified and DNase-treated RNA was reverse transcribed in 20 µl reactions containing 2.5 µg of total RNA or RNA isolated from the input of IPs or ¼ of the RNA isolated from IPs using 5 mg of cell-free extract, 4 µl of 5× Vilo

reaction mix (Life Technologies, 11754), and 2 µl of 2× Superscript III enzyme blend (Life Technologies, 11754). All the reactions were incubated at 25 °C for 10 min, 42 °C for 1 h, and 85 °C for 5 min in a thermocycler. RNase H (5 U, NEB, M0297S) was added followed by incubation at 37 °C for 20 min and 65 °C for 20 min. Reactions were then diluted with water 10-fold for total or input RNA and 20-fold for RNA isolated from IPs. Reactions in 384-well microplates contained 2 µl of the diluted cDNA, 5 µl of 2× PerfeCTa SYBR Green FastMix (Quantabio, 733-2086), 2 µl of forward and reverse primers from stocks at 2.5 µM (listed in Supplementary Table 5), and 1 µl of H$_2$O. The qPCR reactions were performed in technical and biological triplicates in a ViiA 7 Real-Time PCR system (Thermo Fisher Scientific). Microsoft Excel (v16.54) was used to compile the data from the amplification. The qPCR data from total RNA was normalized to three reference genes (act1, his1, and snR101) by the QuantStudio Software V1.2.4 (Thermo Fisher Scientific) while IP RNA was normalized to the input (TER1 all, TER1 precursor). Finally, the relative fold change was calculated by comparing either to an untagged control or to wild type as stated in the figure legends.

**Statistics and reproducibility**. Statistical analysis for all qPCR data was performed on the log$_2$-transformed values of biological triplicates. Each sample was compared with the wild type using two-tailed Dunnett's test from the R package PMCMRplus. Homoscedasticity was tested with Levene's and Bartlett's tests and equal variances for the samples were assumed. In addition, the data were plotted as bar graphs in the original scale along with mean and standard deviation. Significant values in the figures are marked above each bar (***$p < 0.001$, **$0.001 ≤ p < 0.01$, *$0.01 ≤ p < 0.05$). Experiments for Figs. 2a, 3d, and 5a were repeated independently three times. Experiments for Figs. 2b, 4d, f, h, and 6e and Supplementary Figs. 2b–d, 3, 4c, and 5 were repeated independently twice. Experiments for Figs. 2c, 4c, e, g, 5e, and 6c, d, f, g and Supplementary Fig. 4b were performed once. In Fig. 4b, wild type, single deletion, and double deletions were repeated independently four times, and the triple deletion was repeated independently twice.

**Telomerase activity assay**. Telomerase was enriched by IP as described in the section "Immunoprecipitations." Bead suspension aliquots (10 and 20 µl) were used in telomerase activity assays. Control samples were incubated in 20 µl of TMG(50) plus 20 ng of RNase A (Invitrogen) for 10 min at 30 °C prior to the telomerase assay. Supernatants were removed and beads resuspended in 10 µl of 50 mM Tris-acetate at pH 8.0, 100 mM potassium acetate, 5% (v/v) glycerol, 1 mM spermidine, 1 mM DTT, 0.2 mM dATP, dCTP, dTTP, 2 µM [α32P]-dGTP (500 Ci/mmol), and 5 µM of oligo PBoli871 (5′-GTTACGGTTA-CAGGTTACG-3′). Reactions were incubated for 90 min at 30 °C and stopped by the addition of proteinase K (2 µg µl$^{-1}$ in 0.5% [w/v] SDS, 10 mM EDTA pH 8.3, 20 mM Tris-HCl pH 7.5). Following the addition of 1000 counts per minute of a 100-mer oligonucleotide labeled with [γ-$^{32}$P]-ATP as recovery and loading control reactions were incubated at 42 °C for 15 min. DNA was extracted with phenol:-chloroform:isoamyl alcohol (25:24:1, equilibrated with 5× TE) and ethanol precipitated overnight at −20 °C. Products were resuspended in RNA loading dye (NEB, B0363S) and separated by electrophoresis through a 10% (v/v) poly-acrylamide (19:1) sequencing gel containing 8 M urea for 1.5 h at 80 W. Gels were dried exposed to PhosphorImager screens and analyzed with a Typhoon scanner. Telomerase activity was quantified in ImageQuant TL (v7.0) except for Fig. 6g which was quantified by Fiji/ImageJ v1.53c.

**RNA-seq**. Ribosomal RNA was depleted from 2 µg total RNA using Illumina's Ribo-Zero Magnetic Kit for Yeast (Part #15065382 Rev A, November 2014). Library prep was performed with Illumina's TruSeq Stranded Total RNA LT Sample Prep Kit, following the TruSeq Stranded Total RNA Reference Guide (Oct. 2017; Document # 1000000040499v00) and starting with the Elute Fragment Prime High Conc Mix. Libraries were amplified with 8 PCR cycles, profiled with a High Sensitivity DNA Chip on a 2100 Bioanalyzer (Agilent Technologies), and quantified using the Qubit dsDNA HS Assay Kit (Life Technologies). All 12 samples were pooled in equimolar ratio and sequenced on a NextSeq 500 High Output flow cell (1 × 84 cycles, plus 7 cycles for the index read). Read counts per library ranged from 38.36 to 42.88 million. Reads were trimmed and aligned to the S. pombe ASM294v2 genome from ENSEMBL using the STAR aligner (v2.5.2b)[63] with the following parameters: --alignSJDBoverhangMin 5 --alignSJoverhangMin 10 --twopassMode Basic --alignIntronMin 20 --alignIntronMax 2500. Between 34.55 and 41.55 million reads per library passed filtered and aligned uniquely. DESeq2 (v1.22.2) in R was used for differential gene expression analysis[64]. Genes with absolute log$_2$ fold change ≥ 1 and an adjusted p value < 0.05 were considered differentially expressed. Coverage plots of RNA-seq data were generated using pyGenomeTracks[65].

**Telomere length analysis**. DNA preparation and telomere length analysis were performed as described[17]. Cells from 20 ml of stationary culture (~2 × 10$^9$ cells) were incubated with 2 ml of Z buffer (50 mM sodium citrate, 50 mM sodium phosphate dibasic, and 40 mM EDTA pH 7.8) plus 0.5 mg/ml Zymolase T100 (US Biological) and 2 mM DTT for 1 h at 37 °C. SDS was then added to a final concentration of 2% (w/v) and incubated for 10 min at 65 °C. Then 5× TE (50 mM Tris-HCl pH 8.0, 5 mM EDTA) was added to a final volume of 10 ml and

proteinase K (Sigma-Aldrich, P2308) was added to a final concentration of 50 µg ml$^{-1}$. After incubation for 1 h at 50 °C, the samples were precipitated with 3 ml of 5 M potassium acetate for 30 min on ice. The precipitates were removed with two rounds of centrifugation at $3200 \times g$ for 10 min at 4 °C. The supernatant was collected and mixed with 1 volume of 100% isopropanol for 1 h on ice followed by centrifugation at $10,500 \times g$ for 10 min at 4 °C. Genomic DNA was resuspended in 5× TE with 50 µg ml$^{-1}$ RNase A. Resuspended DNA was then incubated for 1 h at 37 °C followed by two rounds of extraction with phenol: chloroform:isoamyl alcohol (25:24:1, equilibrated with 5× TE) and one round of chloroform:isoamyl alcohol (24:1, equilibrated with 5× TE). DNA was ethanol precipitated and resuspended in 1× TE. DNA concentrations were determined on a Qubit 3.0 instrument using the dsDNA BR Assay Kit (Life Technologies, Q32853) and 750 ng of each sample was digested with 40 U of EcoRI (NEB, R0101L) and 2 µl of 10× EcoRI buffer in a 20 µl reaction for 12 h. The digested DNA was separated by electrophoresis on a 1% agarose gel in 0.5× TBE (44.5 mM Tris-borate, 1 mM EDTA at pH 8.3) at 120 V for 6 h. Gels were stained with 1 µg ml$^{-1}$ ethidium bromide and visualized with a Typhoon 8600 scanner to confirm digestion of loaded DNA. Gels were then incubated in 0.25 M hydrochloric acid for 10 min followed by 0.5 M sodium hydroxide and 1.5 M sodium chloride buffer for 30 min and 0.5 M Tris-HCl (pH 7.5) and 1.5 M sodium chloride for 30 min at room temperature. DNA was transferred to Amersham Hybond-N + membrane (GE Healthcare Life Sciences) via capillary blotting. Transferred DNA was crosslinked to the membrane with 120 mJ per cm$^2$ with 254-nm UV light in a Stratalinker. A telomere specific probe was generated by PCR from pTELO using T3 (5′-ATTAA CCCTCACTAAAGGGA-3′) and T7 (5′-TAATACGACTCACTATAGGG-3′) oligos. A probe specific for the *rad16* gene was generated by PCR from wild-type genomic DNA using primers XWP9 (5′-ATGGTATTTTTTCGGCCATT TACTCG-3′) and XWP10 (5′-TAGGCGGATCGTGAAGTTAA-3′). Both probes were labeled by random hexamer labeling with High Prime (Roche, 11585592001) and [α-$^{32}$P]-dCTP. Hybridizations were carried out with 10 million counts per minute of each probe in Church–Gilbert buffer at 65 °C. Blots were exposed to PhosphorImager screens and analyzed with a Typhoon scanner.

**Recombinant protein purification.** Untagged *pof8* and 6xHis N-terminally tagged *thc1* were cloned into the multicistronic vector pET-His6 (9B) (addgene #48284) and transformed into an *E. coli* Rosetta (DE3) pLysS strain. Cultures (500 ml) were grown at 37 °C until an OD$_{600}$ of ~0.5 when the expression was induced by the addition of 1 mM isopropyl β-d-1-thiogalactopyranoside (IPTG), and further incubated for 2 h at 37 °C. Cultures were then harvested by centrifugation once for 20 min at $4000 \times g$ in a Beckman JA-10 rotor, and cell pellet was resuspended in 10 ml lysis buffer (50 mM sodium phosphate buffer pH 8.0, 300 mM NaCl, 10% glycerol (v/v), 10 mM imidazole). Cells were lysed by sonication with 30% amplitude, 15 s on, 15 s off for a total of 1 min 30 s, and lysates were cleared by centrifugation twice for 20 min in a Beckman JS-7.5 at $10,400 \times g$ and JA-17 at $40,000 \times g$ at 4 °C. Cleared lysates were mixed with 0.5 ml Ni-NTA beads (250 µl bed volume, Qiagen), pre-washed with two times 5 ml lysis buffer, in 15 ml tubes and incubated for 30 min at 4 °C with gentle agitation. The lysate-resin mix was transferred to a pre-cooled column and beads were allowed to settle for 15 min prior to the release of flow through. The column was then washed twice with 2.5 ml of wash buffer (50 mM NaH$_2$PO$_4$ buffer pH 8.0, 300 mM NaCl, 40 mM imidazole, 10% (v/v) glycerol). The bound proteins were eluted in 2.5 ml elution buffer (50 mM sodium phosphate buffer pH 8.0, 300 mM NaCl, 10% glycerol, 250 mM imidazole) and collected in 10 fractions of approximately 250 µl each. Finally, the collected fractions were dialyzed against 50 mM sodium phosphate pH 8.0, 150 mM NaCl, 0.5 mM EDTA, 1 mM DTT, 20% glycerol, snap-frozen, and stored at −80 °C.

**Reporting summary.** Further information on research design is available in the Nature Research Reporting Summary linked to this article.

## Data availability

The primary sequence data associated with this analysis have been deposited in NCBI's Gene Expression Omnibus (GEO) database under GEO accession number GSE174822. Raw mass spectrometry data and search results files have been deposited to the Proteome Xchange under accession number PXD026230 and the MassIVE repository ftp://massive.ucsd.edu/MSV000087493/). Original data underlying this manuscript can be accessed from the Stowers Original Data Repository at http://www.stowers.org/research/publications/libpb-1664. Uncropped blots are provided in Supplementary Fig. 7. The data supporting the findings of this study are available from the corresponding authors upon reasonable request. Source data are provided with this paper.

## Code availability

Custom scripts used in this study are available at https://github.com/baumannlab/Sp_Paez_et_al_2021[66].

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

## Acknowledgements

We thank the Stowers Institute for Medical Research Proteomics, Molecular Biology and Media Preparation facilities as well as the Media Lab and Genomics Core Facilities at the Institute of Molecular Biology for excellent service and support as well as the IMB Bioinformatics Core for computing resources and all members of the Baumann laboratory for helpful discussions. The early stages of this work were supported by the Howard Hughes Medical Institute and the Stowers Institute for Medical Research. Funded in part by the Deutsche Forschungsgemeinschaft (DFG, German Research Foundation) – Project-ID 393547839 – SFB 1361. P.B. is an Alexander von Humboldt Professor at Johannes Gutenberg University Mainz.

## Author contributions

D.J.P.-M. and P.B. designed the study; D.J.P.-M. carried out the experiments except for acquisition and primary analysis of the proteomics data (M.J.L. and L.F.), generation of *bmc1* deletion and data related to and presented in Fig. 5a (K.H.), the computational analysis (D.V.H.; Fig. 1c and Supplementary Figs. 2e, f and 6), and data shown in Supplementary Fig. 2g (K.L.J.) and Figs. 4c, e, g, 5c, and 6c–g and Supplementary Fig. 5 (L.P.). All authors contributed to the data analysis and preparation of the manuscript.

## Funding

## Competing interests

The authors declare no competing interests.
