## [Peer Review File · Nature Communications]

Title: A putative cap binding protein and the methyl phosphate capping enzyme Bin3/MePCE function in telomerase biogenesis.REVIEWER COMMENTS

Reviewer #1 (Remarks to the Author):

In this manuscript, the Baumann group reports on two thus far unknown factors involved in the stability and biogenesis of the telomerase RNP in *S. pombe*. Previous studies had already linked the RNA binding protein Pof8 (LARP7) to the loading of the Lsm ring onto the 3' end of TER1 RNA. Using a combination of biochemical and genetic studies, the authors now show that Pof8 acts together with Thc1 and Bmc1 in the biogenesis of telomerase RNP. These uncharacterized proteins in *S. pombe* share homology to the human NCBP3/PARN and MePCE and thus have reported links to snRNAs as well as 7SK RNA.

The results shown in this manuscript are clean and convincing and they contribute to our understanding of the architecture and function of telomerase RNP. The following aspects should be addressed to further support the conclusions of the authors:

The IP shown in Fig. 1 convincingly show that there is a trimeric complex formed by the interaction of Pof8 with Thc1 and Bmc1. The model shown in Fig. 6 suggests that this complex should also contain stoichiometric amounts of the Lsm2-8 ring. The authors should comment on why these factors are underrepresented in their IP. A gradient of the immunoprecipitated material would also inform on the complex composition.

Figure 1 c and d: In lanes 6 and 7 of both blots an unlabeled band of approx. 60 kDa disappears upon RNase treatment. The authors should comment on the identity of this protein and why it disappears under the indicated conditions.

The authors convincingly show that the catalytic activity of Bmc1 is not required in the context of telomerase. I don't think that it is necessary in the context of this paper to show experimentally that Bmc1 has methyltransferase activity. Nevertheless, it would be nice to see an alignment (or even better a homology model) of the catalytic domain of Bmc1 that justifies the claim that Bmc1 indeed is capable of AdoMet binding and performing the methyltransferase reaction.

Minor points:

The "high contrast" images in Fig. 3 d and e can be removed as they don't add anything to the story.

Fig. 4 a and b: The extend of Lsm4 IP should be shown to normalize the effects on TER1 binding and telomerase activity.

There are two recent studies on Larp7 and its link U6 modification that could be discussed in the context of this study.

Reviewer #2 (Remarks to the Author):

In the current manuscript, Paez-Moscoso et al. identified and characterize two additional subunits of fission yeast telomerase complex by affinity purification of FLAG-tagged Pof8 protein. I found that overall findings of their work to be quite exciting, and I think experiments presented were high quality with proper controls, providing convincing supports for their conclusions.

While the manuscript is already in very good quality, I do have a few suggestions/questions as listed below.

1. I am curious that Trt1 was not one of proteins identified mass spec. in supp. Table 1. Did authors not find Trt1 at all, or they did find some peptides for Trt1, but did not meet their cutoff criteria? I am also curious that author did not show any co-IP data for Trt1 with either Thc1 or Bmc1. Were such experiments attempted? If Trt1-Thc1 or Trt1-Bmc1 co-IP is not readily detectable, it is possible that Bin3-Thc1 would not be a stable component of active telomerase as depicted in Fig.6?

2. Along the same line, while authors convincingly showed that Thc1 is required for productive assembly of Lsm4 with catalytically active telomerase (Fig.4) and both Thc1 and Bmc1 promotes association of Pof8 to catalytically active telomerase (Fig.5), there was a curious lack of data for TER1 co-IP or telomerase activity assay utilizing either Thc1 or Bmc1. To fully support a model presented in Fig.6, suggesting Thc1 and Bmc1 as integral components of active telomerase complex, it would be useful to show that Thc1 and Bmc1 is tightly associated with TER1 and can pull down robust telomerase activity.

3. For supp. Table 1, Locus is represented with protein ID, which is not directly searchable in PomBase. Data would be more accessible if authors instead list gene ID numbers, much like in supp. Table 2. I am also not entirely sure how infinity number listed in supplementary Table 1 can be plotted in Figure 1b. I wonder if authors can include dNSAF and log₂ enrichment values used in plotting Figure 1b for individual proteins listed in supplementary Table 1.

4. While I am convinced that authors were indeed able to delete Bmc1 despite being identified as essential gene previously based on supporting data presented, it might be worth mentioning that Bayfield lab has come to conclusion that Bmc1 (Bin3 in their paper) is indeed essential gene in their hand. I am not entirely sure what is the final editorial decision for Bayfield paper, since their paper was considered concurrently, it would be worth mentioning this discrepancy in the final version of manuscript. (Their paper is citable regardless as they uploaded their pre-preint in bioRxiv.) It would also be beneficial if Baumann lab and Bayfield lab can agree on same name for Bmc1/Bin3.

5. The main text describing Bmc1 mutant plasmid expression experiment (supplementary Fig.4) should make it clear experiments are carried out using ectopic exogeneous plasmid.

6. This is only a minor point, but I found it to be bit odd that authors chose to cite their own papers that

identified *ter1* (ref.6) and *pot1* (ref.43) in mentioning that *trt1Δ* cells show telomere loss and chromosome circularization. It seems to me that actual paper that originally identified chromosome circularization in *trt1Δ* (Nakamura et al. Science 1998) should be cited instead in this context?

Reviewer #3 (Remarks to the Author):

In their study Páez-Moscoso et al. identify two novel components, which they named Thc1 and Bmc1, of the *S. pombe* telomerase holoenzyme. Conspicuously, Bmc1 is an orthologue of the methylphosphate capping enzyme (MePCE), which tightly interacts with LARP7 in mammalian cells. Considering that the yeast orthologue of LARP7, Pof8, is a stable component of telomerase, their finding suggests that the association between MePCE and LARP7 is evolutionarily conserved. The authors show that Pof8, Thc1 and Bmc1 interact with each other in an RNA-independent manner, and that deficiency of these factors depletes TER1 RNA. While depletion of Thc1 or Bmc1 reduced the association of Pof8 with TER1, the interaction of Lsm4 as part of the Lsm2-8 complex with TER1 was differentially regulated by Thc1 and Bmc1 indicating that they exert specific functions in telomerase assembly or stability. The manuscript is logically structured and well-written, and the figures are of high quality. The findings are of high significance to the field and extend the current understanding of telomerase composition, implicating novel factors that regulate its biogenesis and stability.

Major Comments:

Additional experiments investigating the association of Thc1 and Bmc1 with core telomerase components would strengthen the view that these factors are part of the mature telomerase holoenzyme. For example, this could be addressed by Trt1 or Lsm4 IP and immunoblot analysis for Thc1, Bmc1 and Pof8. The authors might also consider performing gradient ultracentrifugation to analyze co-sedimentation of Thc1, Bmc1 and Pof8 with other telomerase components. Additionally, telomerase activity assays could be performed with Thc1 and Bmc1 IP samples.

Given that mammalian MePCE and LARP7 interact directly, it would be of interest to test such direct interaction between Pof8 and Bmc1 using, for example, recombinant proteins. The results shown in Fig. 1c do not rule out the possibility that additional proteins present in the extract mediate the interaction.

Fig. 3c-e: The authors show that deletion of *bmc1*, *thc1* or *pof8* has strong effects on the levels of the respective other protein components of the Thc1-Bmc1-Pof8 complex. In $\Delta bmc1$ cells, Thc1 and, to a lesser extent, Pof8 are reduced (Fig. 3c, input lanes). In $\Delta thc1$ cells, Bmc1 is reduced (Fig. 3d). In $\Delta pof8$ cells, Bmc1 is reduced (Fig. 3e). A quantitative immunoblot analysis of Thc1, Bmc1 and Pof8 protein levels in these deletion strains might greatly substantiate this finding and could precede the figures showing the IPs. Additionally, are the reductions in protein levels due to reduced mRNA amounts?

Fig. 5b: The authors show that Bmc1 is required for Pof8 association with TER1. This seemingly contrasts with the result in Fig. 4a showing that Lsm4 binding to TER1 is unaffected in $\Delta bmc1$ cells, and with their

previous study (Nat Comm 2018) showing that Pof8 is necessary for Lsm2-8 association with TER1. The authors suggest that Bmc1 has a stabilizing rather than assembly function. Can the authors provide data to support this hypothesis? One way might be to investigate the association of Pof8 with Lsm4 (or Trt1) in $\Delta bmc1$ cells. The prediction would be that Lsm2-8 association with TER1 progresses normally in $\Delta bmc1$ cells (as shown in Fig. 4a) but that Pof8 is lost from mature telomerases subsequently.

Minor Comments:

Throughout the text, figure panels are sometimes in capital and sometimes in lowercase letters (e.g. line 224 “Figure 2A”, line 233 “Figure 2b”).

Line 224: “others”

Line 354: “dependent on”

Lines 347/348: Should be lane 4.

Line 1012/13: In Fig. 3e Thc1 was immunoprecipitated but in the legend it says that anti-Strep-tag II antibody was used for IP. Since Thc1 is tagged with V5 I guess the authors used anti-V5 antibody for IP.

Line 1039: “(individual values shown as dots)” should be in the legend of Fig. 2a (first occurrence)

Line 1040: “compromised in”

RESPONSE TO REVIEWER COMMENTS

We thank the three reviewers for their thoughtful and constructive comments. The additional experiments carried out and modifications to the text made in response to the questions raised by the reviewers have made this a clearer and stronger manuscript. Below we respond to each specific comment. Modifications to the text made in response to reviewer comments are shown in blue. New data added to the manuscript is now shown in F1c, SF4a, F4c, e, g, F5c, SF5, F6c, d, e, f and g. Due to the additional data, the numbering of other panels deviates from the original submission.

Reviewer #1

The IP shown in Fig. 1 convincingly show that there is a trimeric complex formed by the interaction of Pof8 with Thc1 and Bmc1. The model shown in Fig. 6 suggests that this complex should also contain stoichiometric amounts of the Lsm2-8 ring. The authors should comment on why these factors are underrepresented in their IP. A gradient of the immunoprecipitated material would also inform on the complex composition.

Based on prior work by the Nakamura, Tomita, Qiao and our lab as well as the data presented here, we agree that Pof8, Thc1, Bmc1 and each of the Lsm2-8 proteins are likely present in a 1:1 stoichiometry. However, such a complex does not appear to be very stable. We note that a substantial fraction of the Flag-tagged Pof8 protein appears to be associating with heat-shock protein Ssa2 in the IP (Suppl. Figure 1a and b) and based on Silver stained gels and dNSAF values even Thc1 and Bmc1 are not present in stoichiometric amounts in the Pof8 immunoprecipitate. Five of the seven Lsm proteins were detected, but with much lower dNSAF values than Pof8. This may in part be due to these being small proteins which are more difficult to detect in label-free quantitative MS. But it could also be a consequence of only a fraction of Pof8 being in complex with Thc1, Bmc1 and an intact Lsm2-8 ring and such a complex not being very stable under the IP conditions. The fact that Bmc1 is largely dispensible for the loading of Lsm2-8, but not for stable association of Pof8 with Trt1/TER1 further supports a dynamic nature of this complex. This is now explicitly discussed in the text also in light of the new data shown in Figures 6c to g and Suppl Figures 5. Attempts to further characterize the complex composition from Pof8 immunoprecipitations has not led to additional insights thus far.

Figure 1 c and d: In lanes 6 and 7 of both blots an unlabeled band of approx. 60 kDa disappears upon RNase treatment. The authors should comment on the identity of this protein and why it disappears under the indicated conditions.

Comment now refers to Figure 2a and b in revised version. The 60 kDa band is present in anti-FLAG immunoprecipitates even when no FLAG-tagged protein is present in the cells from which the extract was prepared (see for example Figure 2a, lane 3). It is therefore a fission yeast protein that cross-reacts with the anti-FLAG antibody used here independent of the introduction of the epitope tag. Based on the disappearance of this band from the IP when treated with RNaseA or benzonase, we surmise that this protein of unknown identity associates in an RNA-dependent manner with the anti-FLAG beads. However, as the association is independent of the presence of the FLAG-tag on Pof8, we did not investigate the identity of the 60 kDa protein. The band is now labelled with an asterisk in the figures 2a and b), and the fact that the presence of the band is independent of FLAG-Pof8 is stated in the legend.

The authors convincingly show that the catalytic activity of Bmc1 is not required in the context of telomerase. I don't think that it is necessary in the context of this paper to show experimentally that

Bmc1 has methyltransferase activity. Nevertheless, it would be nice to see an alignment (or even better a homology model) of the catalytic domain of Bmc1 that justifies the claim that Bmc1 indeed is capable of AdoMet binding and performing the methyltransferase reaction.

We have replaced the original Supplementary Figure 1c with a multi-sequence alignment and homology model (new Figure 1c) and present a Chimera-X overlay of the Bmc1 structure predicted by Alphafold with the solved structure of human MePCE in complex with S-adenosylhomocysteine (New Suppl Figure 4a). Both show clear structural conservation and compatibility of the *S. pombe* sequence with catalytic activity.

Minor points:

The “high contrast” images in Fig. 3 d and e can be removed as they don’t add anything to the story.

These panel have been removed.

Fig. 4 a and b: The extend of Lsm4 IP should be shown to normalize the effects on TER1 binding and telomerase activity.

This comment now refers to Figure 5d and e. We have added Western blots showing that the levels of Lsm4 in extracts and c-Myc immunoprecipitates are the same in the different genetic backgrounds.(10% variation lies within technical limitations of the experiment) This data is now shown in the new Panel 5c.

There are two recent studies on Larp7 and its link U6 modification that could be discussed in the context of this study.

These studies are now mentioned in the introduction and discussion sections.

Reviewer #2 (Remarks to the Author):

1. I am curious that Trt1 was not one of proteins identified mass spec. in supp. Table 1. Did authors not find Trt1 at all, or they did find some peptides for Trt1, but did not meet their cutoff criteria? I am also curious that author did not show any co-IP data for Trt1 with either Thc1 or Bmc1. Were such experiments attempted? If Trt1-Thc1 or Trt1-Bmc1 co-IP is not readily detectable, it is possible that Bin3-Thc1 would not be a stable component of active telomerase as depicted in Fig.6?

Although Trt1 peptides were not detected in our proteomics experiments, we did show that the RNA subunit TER1 is strongly enriched in Thc1 and Bmc1 immunoprecipitates (Figure 2c, now 3c). As this does not imply an association with the catalytic subunit, we have now included data showing telomerase activity associated Thc1 and Bmc1 immunoprecipitates (new Suppl. Figure 5). Due to the myc-tag on Trt1 affecting its activity, we decided to use telomerase activity as read-out rather than just co-IP Western.

2. Along the same line, while authors convincingly showed that Thc1 is required for productive assembly of Lsm4 with catalytically active telomerase (Fig.4) and both Thc1 and Bmc1 promotes

association of Pof8 to catalytically active telomerase (Fig.5), there was a curious lack of data for TER1 co-IP or telomerase activity assay utilizing either Thc1 or Bmc1. To fully support a model presented in Fig.6, suggesting Thc1 and Bmc1 as integral components of active telomerase complex, it would be useful to show that Thc1 and Bmc1 is tightly associated with TER1 and can pull down robust telomerase activity.

We thank reviewers 2 and 3 for similar comments on this topic and agree that this was a logical extension of the work presented in the original manuscript. We have now included telomerase activity assays from Thc1 and Bmc1 immunoprecipitates and found that Thc1 precipitates contain similar levels of telomerase activity to Pof8 immunoprecipitates, whereas Bmc1 immunoprecipitates contain less telomerase activity consistent with the lower amount of TER1 recovered from such immunoprecipitates (shown in Figure 3c and Supp Figure 5). It is important to note the limitations of this experiment though, as differences between tags and differences in the effect of any tag on the activity of a specific protein prohibit direct quantitative comparison of the associated telomerase activity. For example, we found previously that Lsm proteins precipitate more telomerase activity than Trt1 itself (when normalized by the amount of recovered TER1). After an initial excitement over the possibility that we are pulling down a more active fraction of telomerase with Lsm, we came to the conclusion that epitope tags on Trt1 reduce its activity relative to untagged Trt1. Therefore, the telomerase assay for Bmc1 and Thc1 precipitates only show that the activity level tracks the amount of co-precipitating TER1 (also compared to Pof8). To take this one step further we asked what fraction of telomerase activity is associated with Thc1 and Bmc1 through sequential immunoprecipitation and activity measurements. We first showed that over 90% of Thc1 and Bmc1 are respectively removed from extracts in a single step IP (immunodepletion; new Figure 6e and f). We then used the supernatant of these Thc1 and Bmc1 IPs and performed a second round of IP against Lsm4-cMyc, previously shown to be stably associated with active telomerase (Tang et al., Nature 2012). Comparing activity between first and second round IPs allowed us to assess what fraction of active telomerase is associated with Thc1 and Bmc1 respectively. This data has now been included in the manuscript as new Figure 6g.

3. For supp. Table 1, Locus is represented with protein ID, which is not directly searchable in PomBase. Data would be more accessible if authors instead list gene ID numbers, much like in supp. Table 2. I am also not entirely sure how infinity number listed in supplementary Table 1 can be plotted in Figure 1b. I wonder if authors can include dNSAF and log2 enrichment values used in plotting Figure 1b for individual proteins listed in supplementary Table 1.

Suppl. Table 1 has now been changed accordingly to list gene IDs instead of protein IDs. Infinity values for fold enrichment arise when zero peptides are detected in the control samples. To plot these values at the right edge of the graph in Figure 1b, the enrichment value is arbitrarily set to 2^{15} . This is now detailed in the Figure legend.

4. While I am convinced that authors were indeed able to delete Bmc1 despite being identified as essential gene previously based on supporting data presented, it might be worth mentioning that Bayfield lab has come to conclusion that Bmc1 (Bin3 in their paper) is indeed essential gene in their hand. I am not entirely sure what is the final editorial decision for Bayfield paper, since their paper was considered concurrently, it would be worth mentioning this discrepancy in the final version of manuscript. (Their paper is citable regardless as they uploaded their pre-print in bioRxiv.) It would also be beneficial if Baumann lab and Bayfield lab can agree on same name for Bmc1/Bin3.

I have discussed these issues with Dr. Bayfield and offered to exchange reagents and strains to resolve the issue of viability of *bmc1Δ* cells. It is my understanding that his lab is making further attempts to delete the gene prior to resubmission. Dr. Bayfield and Valerie Wood at Sanger agreed that *Bmc1* should be the preferred choice for the official gene name for the reasons we give in the manuscript. As several fungal orthologs are already found under the name *Bin3* in various databases, they both suggested and I agreed to keep listing the name *Bin3* as a synonym. We are now referencing the BioRxiv preprint in our manuscript.

5. The main text describing *Bmc1* mutant plasmid expression experiment (supplementary Fig.4) should make it clear experiments are carried out using ectopic exogenous plasmid.

The text has been changed to make this clear.

6. This is only a minor point, but I found it to be bit odd that authors chose to cite their own papers that identified *ter1* (ref.6) and *pot1* (ref.43) in mentioning that *trt1Δ* cells show telomere loss and chromosome circularization. It seems to me that actual paper that originally identified chromosome circularization in *trt1Δ* (Nakamura et al. Science 1998) should be cited instead in this context?

Clearly our mistake, which at this point nobody has an explanation for. The correct reference is now cited.

Reviewer #3 (Remarks to the Author):

Major Comments:

Additional experiments investigating the association of *Thc1* and *Bmc1* with core telomerase components would strengthen the view that these factors are part of the mature telomerase holoenzyme. For example, this could be addressed by *Trt1* or *Lsm4* IP and immunoblot analysis for *Thc1*, *Bmc1* and *Pof8*. The authors might also consider performing gradient ultracentrifugation to analyze co-sedimentation of *Thc1*, *Bmc1* and *Pof8* with other telomerase components. Additionally, telomerase activity assays could be performed with *Thc1* and *Bmc1* IP samples.

To further characterize the association of *Thc1* and *Bmc1* with core telomerase components, we performed co-IP analysis. *Thc1* and *Bmc1* are readily co-immunoprecipitated with *Lsm4* (new Figure 6c and d). We used telomerase activity assays as read out for the presence of the catalytic subunit. Telomerase activity from *Bmc1* and *Thc1* immunoprecipitations is now shown as new Suppl Figure 5. To assess what fraction of telomerase activity is associated with *Thc1* and *Bmc1* respectively, we also performed sequential IPs as previously used for *Pof8*. This data is now shown in new Figure 6e, f and g. See also response to reviewer 2, points 1 and 2.

Given that mammalian MePCE and LARP7 interact directly, it would be of interest to test such direct interaction between *Pof8* and *Bmc1* using, for example, recombinant proteins. The results shown in Fig. 1c do not rule out the possibility that additional proteins present in the extract mediate the interaction.

At this stage we cannot rule out the possibility that the interaction between *Bmc1* and *Pof8* is indirect and we now state this in the text on page 12. We have made numerous attempts to purify

recombinant Bmc1 by itself or in combination with Pof8 or Thc1 thus far without success. With the first and fourth author having left the lab, we have a new student on the purification project, but we hope that the reviewer agrees that further characterization of the Bmc1/Pof8 interaction can be the subject of a future manuscript.

Fig. 3c-e: The authors show that deletion of bmc1, thc1 or pof8 has strong effects on the levels of the respective other protein components of the Thc1-Bmc1-Pof8 complex. In $\Delta bmc1$ cells, Thc1 and, to a lesser extent, Pof8 are reduced (Fig. 3c, input lanes). In $\Delta thc1$ cells, Bmc1 is reduced (Fig. 3d). In $\Delta pof8$ cells, Bmc1 is reduced (Fig. 3e). A quantitative immunoblot analysis of Thc1, Bmc1 and Pof8 protein levels in these deletion strains might greatly substantiate this finding and could precede the figures showing the IPs. Additionally, are the reductions in protein levels due to reduced mRNA amounts?

We performed quantitative immunoblots to better assess the extent to which deletion of bmc1, thc1 or pof8, respectively affects the steady-state level of the other two components. This data is now included as new Figure 4c, e and g. We also examined the RNAseq data from pof8 Δ , thc1 Δ and bmc1 Δ strains and detected no significant reduction in mRNA levels of the remaining two factors, suggesting that the reduction is a matter of protein stability not gene expression.

Fig. 5b: The authors show that Bmc1 is required for Pof8 association with TER1. This seemingly contrasts with the result in Fig. 4a showing that Lsm4 binding to TER1 is unaffected in $\Delta bmc1$ cells, and with their previous study (Nat Comm 2018) showing that Pof8 is necessary for Lsm2-8 association with TER1. The authors suggest that Bmc1 has a stabilizing rather than assembly function. Can the authors provide data to support this hypothesis? One way might be to investigate the association of Pof8 with Lsm4 (or Trt1) in $\Delta bmc1$ cells. The prediction would be that Lsm2-8 association with TER1 progresses normally in $\Delta bmc1$ cells (as shown in Fig. 4a) but that Pof8 is lost from mature telomerases subsequently.

In the absence of Bmc1, the steady-state association between Pof8 and TER1 is indeed 4 fold reduced, whereas the association between Lsm4 and TER1 is unchanged or slightly increased. As the reviewer suggests, Lsm2-8 association with TER1 must proceed normally in the absence of bmc1 based on in vitro activity levels in Lsm4 IPs (Figure 5e). Furthermore, based on telomere length, Lsm2-8 and Trt1 association with TER1 must also proceed normally in cells (or close to normally) as telomeres are only slightly shorter in bmc1 Δ cells compared to wildtype (Figure 3d). In light of these results, it was indeed surprising to see that stable association of telomerase with Pof8 is strongly reduced in the absence of Bmc1. We concluded that Pof8 must be able to fulfill its function in helping Lsm2-8 load, even though its association with Trt1/TER1 is significantly less stable when Bmc1 is missing.

Minor Comments:

Throughout the text, figure panels are sometimes in capital and sometimes in lowercase letters (e.g. line 224 "Figure 2A", line 233 "Figure 2b").

This has been fixed

Line 224: "others"

Fixed

Line 354: "dependent on"

Fixed

Lines 347/348: Should be lane 4.

Fixed

Line 1012/13: In Fig. 3e Thc1 was immunoprecipitated but in the legend it says that anti-Strep-tag II antibody was used for IP. Since Thc1 is tagged with V5 I guess the authors used anti-V5 antibody for IP.

This error has been corrected.

Line 1039: "(individual values shown as dots)" should be in the legend of Fig. 2a (first occurrence)

Inserted accordingly

Line 1040: "compromised in"

fixed

REVIEWERS' COMMENTS

Reviewer #1 (Remarks to the Author):

The authors have addressed the comments/criticisms of the referees in a satisfactory manner and they included additional experiments and controls that have improved the work considerably. I therefore support publication of this interesting study in Natcomm.

Reviewer #2 (Remarks to the Author):

I am satisfied with responses and changes made to the manuscript by authors. I think data in paper is very robust and provide important new insights to telomerase regulation. I enjoyed reading the manuscript. Thank you.

Reviewer #3 (Remarks to the Author):

The authors have responded adequately to my previous comments.

Minor comment:

Page 17, line 394: there is no separate panel h in Figure 6

Response to reviewers for revised version of manuscript NCOMMS-21-18384A

REVIEWERS' COMMENTS

Reviewer #1 (Remarks to the Author):

The authors have addressed the comments/criticisms of the referees in a satisfactory manner and they included additional experiments and controls that have improved the work considerably. I therefore support publication of this interesting study in Natcomm.

Response: We thank Reviewer 1 for his/her constructive comments on the initial submission and their appreciation of our revisions. The comments helped us to improve the manuscript considerably.

Reviewer #2 (Remarks to the Author):

I am satisfied with responses and changes made to the manuscript by authors. I think data in paper is very robust and provide important new insights to telomerase regulation. I enjoyed reading the manuscript. Thank you.

Response: We thank Reviewer 2 for his/her constructive comments on the initial submission and their appreciation of our revisions. The comments helped us to improve the manuscript considerably.

Reviewer #3 (Remarks to the Author):

The authors have responded adequately to my previous comments.

Minor comment:

Page 17, line 394: there is no separate panel h in Figure 6

Response: We thank Reviewer 3 for his/her constructive comments on the initial submission and the revisions. The incorrect reference to panel 6h has been corrected.